# Latent Thinking Optimization: Your Latent Reasoning Language Model Secretly Encodes Reward Signals in Its Latent Thoughts

**Hanwen Du**[♣] **Yuxin Dong**[♣] **Xia Ning**[♣♠♡✉]

[♣]Department of Computer Science and Engineering, The Ohio State University, USA
[♠]Department of Biomedical Informatics, The Ohio State University, USA
[♡]Translational Data Analytics Institute, The Ohio State University, USA
`{du.1128, dong.1357, ning.104}@osu.edu`

## Abstract

Large Language Models (LLMs) excel at problem solving by generating chain of thoughts in natural language, but such *verbal thinking* is computationally costly and prone to overthinking. A recent work instead proposes a *latent thinking* architecture, Huginn-3.5B, which represents intermediate reasoning steps as a sequence of latent representations. However, latent thoughts lack interpretability and are difficult to supervise, raising concerns about the correctness and reliability of the model's latent thinking processes. In this paper, we provide a systematic study of how Huginn-3.5B thinks in the latent space and how external supervision signals can improve its latent thinking processes. We show that latent thoughts leading to correct versus incorrect answers exhibit highly distinguishable patterns, and that a latent classifier can reliably predict answer correctness directly from latent thoughts. Leveraging these insights, we propose Latent Thinking Optimization (LTO), a probabilistic algorithm that employs the latent classifier as a Latent Reward Model (LRM) to optimize the latent thinking processes. Extensive experiments across diverse reasoning tasks demonstrate that LRM is highly effective in detecting incorrect latent thinking patterns, and LTO can significantly improve the latent thinking processes. Furthermore, we show that LRM can generalize across diverse domains, and LTO can be seamlessly applied to general LLMs to improve their thinking processes. In contrast to verbal thinking, our method demonstrates that reward modeling and scaling test-time thinking with supervision can be performed directly in the latent space, highlighting its potential as a general, efficient, and domain-agnostic approach to improving the thinking processes of LLMs.

## 1 Introduction

Large Language Models (LLMs) (OpenAI, 2023; Bai et al., 2023; Touvron et al., 2023a; Gemini Team, 2023) have demonstrated impressive problem-solving abilities by generating natural language as a form of thinking and reasoning[1] (Wei et al., 2022; Kojima et al., 2022; Yao et al., 2023). This ability to "think" enables them to solve a variety of complex tasks, such as math (Lightman et al., 2024; Gao et al., 2023), coding (Li et al., 2022; Nijkamp et al., 2023), and embodied planning (Shinn et al., 2023; Hao et al., 2023). However, generating the whole thinking process in natural language is very costly and prone to the overthinking issue where LLMs output redundant or misleading thoughts that degrade both accuracy and efficiency (Sui et al., 2025; Chen et al., 2025).

In contrast, humans think largely through internal latent representations—compact, abstract mental codes that capture abstract concepts and hidden structures (Quiroga et al., 2005; Mishchanchuk et al., 2024). Such a latent thinking process is highly efficient as it avoids the need to verbalize every intermediate step, and is well-suited for reasoning with abstract logic or concepts that are often difficult to convey through natural language. Motivated by this, a recent work explores modeling

---

[1]In this paper, we use the terms "thinking" and "reasoning" interchangeably to refer to the process by which an LLM generates intermediate steps or latent thoughts toward an answer.

the thinking process as a sequence of latent representations (i.e., latent thoughts) and proposes a new latent reasoning language model Huginn-3.5B (Geiping et al., 2025), where each latent thought corresponds to a thinking step. These latent thoughts form a latent reasoning chain that enable the model to reason effectively in the latent space and achieve impressive performance across a variety of reasoning tasks.

Despite promising, such latent thinking architecture faces a major challenge: it lacks interpretability and supervision. Unlike verbal thinking, where each intermediate step can be inspected and evaluated (Wang et al., 2024), latent thinking is encoded in internal hidden states that are hard to interpret. This makes it difficult to understand what the model is actually thinking about or to verify its correctness. Furthermore, the model is trained to generate these latent thoughts in an unsupervised manner without explicit supervision or reward signals that can indicate what a "good" latent thought is. This raises concerns on whether the model is truly learning to think in the latent space, or simply memorizing the answers using the parameters of latent representations (Wang et al., 2025b).

In this paper, we aim to understand how Huginn-3.5B thinks in the latent space and how external supervision signals can improve its latent thinking process. Specifically, we observe that latent thinking trajectories (i.e., sequences of latent thoughts) that lead to correct versus incorrect answers exhibit distinct patterns. To further investigate this, we train a latent classifier to predict answer correctness from the latent thinking trajectories, and observe that it can reliably distinguish between correct and incorrect trajectories, even for partial trajectories with just the first few thinking steps.

Building on these insights, we formulate latent thinking improvement as a reward optimization problem over latent policies, and propose a Latent Thinking Optimization (LTO) algorithm that uses the latent classifier as a Latent Reward Model (LRM) to sample latent thinking trajectories with a higher estimated likelihood of correctness. LTO is theoretically guaranteed to improve the expected correctness rate and empirically yields significant gains across a range of challenging reasoning tasks.

While we use Huginn-3.5B as a starting point to understand the latent thinking processes, the proposed LRM and LTO extend naturally to general LLMs. Although general LLMs do not explicitly incorporate latent thinking, their latent representations across multiple layers can be interpreted as latent chain of thoughts (Wang et al., 2025c). Under this view, LRM and LTO can be readily applied to general LLMs. In our experiments, we demonstrate that the latent thoughts from general LLMs also encode appropriate reward signals and LTO can significantly improve the performance of general LLMs on diverse reasoning tasks using these LRMs. Furthermore, we show that LRM exhibits strong cross-domain generalization even with a small amount of training data, highlighting its potential as an efficient and generalist reward model in the latent space. In contrast to verbal thinking approaches that scale test-time compute through natural language generation (Guo et al., 2025; Muennighoff et al., 2025), our method demonstrates that reward modeling and scaling test-time thinking with supervision can be performed directly in the latent space, highlighting its potential as a general, efficient, and domain-agnostic approach to improving the thinking processes of LLMs.

## 2 DEFINITIONS AND NOTATIONS OF REASONING LLMs

Given a question $x \sim \mathcal{D}$ sampled from the dataset $\mathcal{D}$, a language model $\pi(\cdot)$ can directly generate an answer by sampling from $y \sim \pi(y \mid x)$. For complex questions, however, it is often beneficial to introduce intermediate reasoning steps $z$ to represent the model's thinking process. In this case, the model first thinks by sampling from $z \sim \pi(z \mid x)$ and then generates the final answer conditioned on $z$, that is, $y \sim \pi(y \mid z)$. Empirically, this two-stage generation process often improves the answer correctness rate, as generating $z$ allows the model to decompose a complex problem into simpler subproblems that are easier to solve and verify (Wei et al., 2022; Kojima et al., 2022).

**Verbal Thinking**   Common reasoning LLMs (Zhang et al., 2026) represent $z$ as a sequence of reasoning steps (Wei et al., 2022) in natural language, that is, $z = (e_1, \cdots, e_t, \cdots, e_T)$, where each $e_t$ is a chunk of text that corresponds to a specific step in the reasoning process. However, generating all the reasoning steps in natural language introduces significant computational overhead, and increases the risk of overthinking where the model generates unnecessarily verbose or logically inconsistent reasoning chains that lead to incorrect answers (Chen et al., 2025; Sui et al., 2025).

**Latent Thinking**   To address the limitations of verbal thinking, inspired by the human cognitive theory, a recent work proposes a latent reasoning language model Huginn-3.5B (Geiping et al.,

2025) which represents the sequence of reasoning steps as a sequence of internal hidden states $z = (\mathbf{h}_1, \cdots, \mathbf{h}_t, \cdots, \mathbf{h}_T)$ (i.e., a latent thinking trajectory). Each $\mathbf{h}_t \in \mathbb{R}^{L \times d}$ represents a latent reasoning step (i.e., a latent thought), where $L$ is the number of tokens in the output $y$, $d$ is the hidden dimensionality. The number of thinking steps $T$ is set as 32 by default, and can vary according to the computation budget. The initial latent thought $\mathbf{h}_0 \sim \mathcal{N}(\mathbf{0}, \sigma^2 \mathbb{I}_{L \cdot d})$ is sampled from random Gaussian noise with the standard deviation $\sigma$, and a recurrent block is introduced to generate the latent thoughts $\mathbf{h}_{1:T}$ recursively conditioned on the question $x$. A lightweight decoding module generates the answer $y$ in natural language conditioned on the last latent thought $\mathbf{h}_T$. Because both chain-of-thought reasoning and recurrent architectures can be conceptualized as finite-state automata (Svete & Cotterell, 2023; Zhang et al., 2024), this approach can be viewed as generating the chain of thoughts in the latent space without the need for verbose reasoning. While efficient, it is difficult to trace the model's logic or provide step-level supervision due to the lack of interpretable patterns in the latent space.

## 3 DECIPHER HOW HUGINN-3.5B THINKS IN THE LATENT SPACE

Latent thoughts are hidden states and may not have an intrinsic notion of "correctness" themselves. To determine what constitutes a "good" or "bad" latent thought, in this paper, we define the correctness of a latent thinking trajectory (latent thinking process) in terms of whether the trajectory (thinking process) leads to a correct answer. This definition provides a reference point for distinguishing "good" from "bad" latent thoughts and enables us to systematically investigate whether these trajectories exhibit distinct patterns in latent space. It is also consistent with the idea of process reward models (Wang et al., 2024; Lu et al., 2024), where the correctness of intermediate reasoning steps is labeled based on their relation with the final answer.

To understand the latent thinking processes, we consider an interesting research question:

> **Research Question:** Do latent thoughts that lead to correct answers exhibit different patterns in latent space compared to those leading to incorrect answers?

If differences exist, they would not only provide insights into how Huginn-3.5B encodes abstract concepts during its thinking process, but also provide a foundation for detecting and correcting thinking errors directly in the latent space.

### 3.1 VISUALIZATION OF LATENT THOUGHTS

To answer this research question, we select two datasets from different domains: SVAMP (Patel et al., 2021) (grade school math) and MBPP (Austin et al., 2021) (python programming). For each problem in these datasets, we randomly sample 100 latent thinking trajectories by sampling from initial latent thought $\mathbf{h}_0$ with different random seeds, and generate the corresponding answers from these latent thoughts. To compare the difference between latent thoughts that lead to correct and incorrect answers, we select those problems that contain both correct and incorrect answers, then visualize and compare their latent thoughts in Figure 1. We have the following observations:

**Correct and incorrect latent thoughts exhibit different structures in the latent space.** For the same problem, the trajectories of correct and incorrect latent thoughts diverge in both their paths and endpoints, indicating that the model is engaging in different thinking behaviors for correct and incorrect solutions. Interestingly, the distributions of correct latent thoughts are relatively compact and tend to converge toward consistent solution paths. In contrast, incorrect latent thoughts are more dispersed in the latent space, suggesting that they lack a stable and consistent reasoning pattern.

**Both correct and incorrect latent thoughts exhibit different thinking dynamics at different steps.** Latent thoughts from early steps show sharp and abrupt changes. This suggests that the model is probably doing active computation and exploratory reasoning, which might involve cognitive behaviors such as searching or backtracking (Gandhi et al., 2025) that are helpful for problem solving. Latent thoughts evolve more smoothly in the middle steps, suggesting that the model is probably finetuning its thinking process for iterative refinement (Madaan et al., 2023). In the last few steps, latent thoughts almost converge, indicating that the thinking process is complete and a

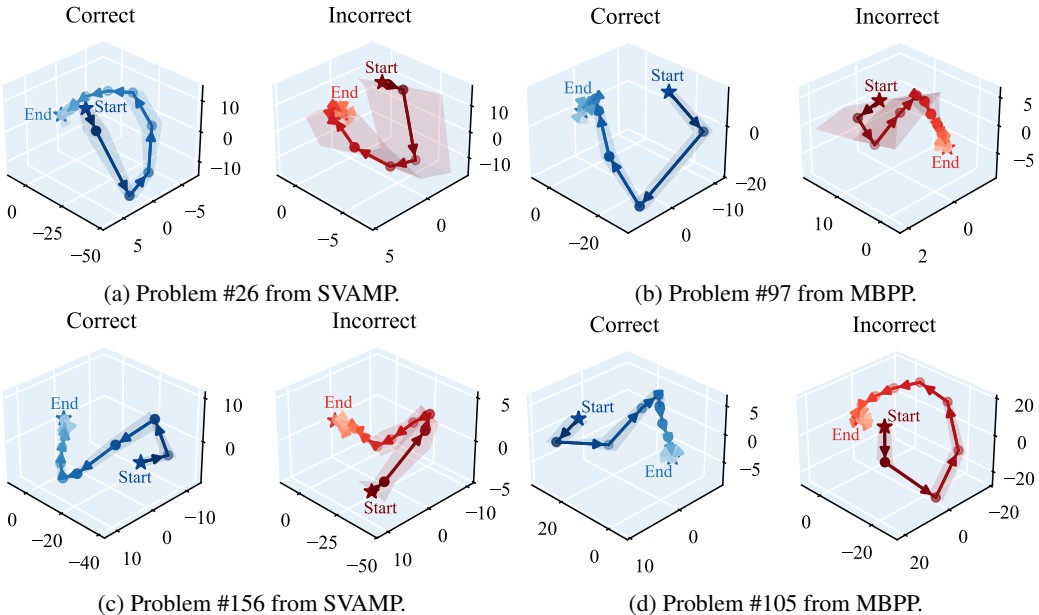

Figure 1: Visualization of the distribution of the correct and incorrect latent thoughts projected onto 3D space using PCA for dimension reduction. The arrows along the lines indicate the progression from the current step to the next step of the latent thought. More examples are in Appendix B.

conclusion is reached. These patterns suggest that the latent space effectively captures the progression of the thinking dynamics, with distinct behaviors emerging at different steps.

**Distinct thinking patterns emerge for different types of problems.** Latent thoughts from math problems display different thinking patterns from those observed in programming problems. Within the same dataset, the model also generates latent thoughts with different patterns for different types of problems. Notably, convergence of latent thoughts is faster on math problems, which typically require only two to three arithmetic computations (Patel et al., 2021). By comparison, the latent thoughts take more steps to converge for programming problems, which are more difficult and require longer reasoning steps (Austin et al., 2021). These observations indicate that the model can flexibly adjust its thinking strategy in response to different problem types and difficulty levels.

## 3.2 QUALITATIVE AND QUANTITATIVE ANALYSES ON LATENT THOUGHTS

The case studies in Section 3.1 demonstrate that the model is engaging in different thinking behaviors for latent thoughts that lead to correct and incorrect answers. To gain a deeper understanding of its thinking processes, we evaluate the latent thoughts at different thinking steps using four metrics that measure the quality of latent representations from the perspective of information content (Entropy, Effective Rank) and geometric structure (Anisotropy, Intrinsic Dimension). These metrics are calculated using all the samples from each dataset.

- **Entropy** (Skean et al., 2025) quantifies how much information content the latent representations carry. A higher entropy indicates the latent representations contain diverse, more informative features, while a lower entropy suggests the existence of redundant information.
- **Effective Rank** (Wei et al., 2024) measures how the dimensionality of the latent representation effectively shrinks under strong compression. A higher effective rank implies noisy features, while a lower effective rank indicates better noise reduction and more compact representations.
- **Anisotropy** (Razzhigaev et al., 2024) measures the non-uniformity of a distribution in the latent space. A higher anisotropy suggests that representations are more directed in specific orientations, while a lower anisotropy indicates that the representations are spread out more evenly.
- **Intrinsic Dimension** (Facco et al., 2017; Cheng et al., 2025) quantifies the minimal number of coordinates required to describe the local geometric structure of the representations without sig-

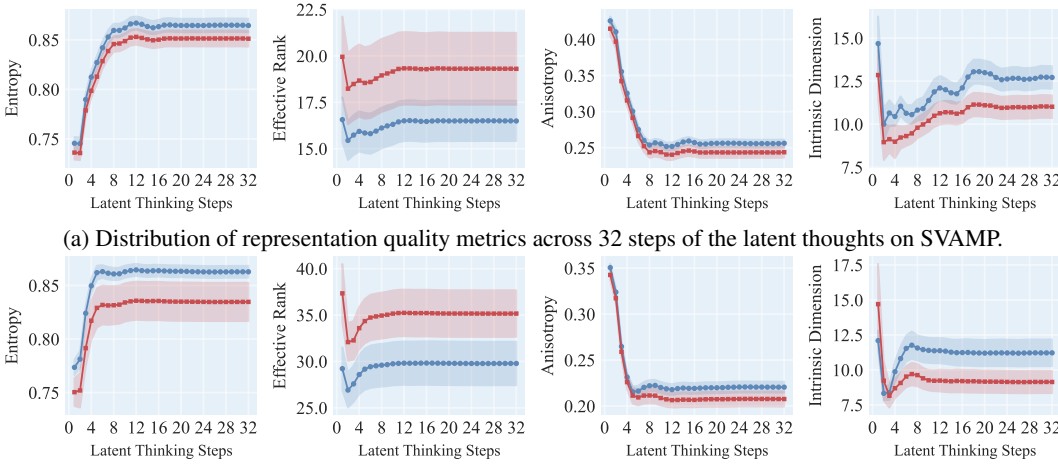

(a) Distribution of representation quality metrics across 32 steps of the latent thoughts on SVAMP.

(b) Distribution of representation quality metrics across 32 steps of the latent thoughts on MBPP.

Figure 2: Representation quality metrics of the latent thoughts on two datasets. The blue and red distributions represent the distributions for the correct and incorrect trajectory of latent thoughts, respectively. These metrics are calculated using all the samples from each dataset.

nificant information loss. A higher intrinsic dimension indicates a rich, complex latent structure, while a lower intrinsic dimension suggests the representation lies on a simpler manifold.

The calculation details of these metrics are in Appendix C. From the visualization of the representation quality metrics across all the thinking steps in Figure 2, we have the following observations:

**Correct thinking processes carry richer information with less noise.** The entropy of correct latent thoughts is consistently higher than that of incorrect ones, and the effective rank of correct latent thoughts is consistently lower. This suggests that correct thinking processes can preserve richer and more informative features (higher entropy), while reducing noisy components (lower effective rank). These observations are consistent with the view of language modeling as a form of compression (Deletang et al., 2024), where effective thinking of LLMs can be understood as a process of extracting the key concepts while discarding noisy or redundant information.

**Correct thinking processes generate more expressive latent representations with structured and complex geometries.** The anisotropy and the intrinsic dimension of correct latent thoughts are consistently higher than those of incorrect ones. This suggests that correct latent thoughts align well along informative directions in the latent space, with a richer and more diverse manifold structure capable of capturing task-relevant features (Valeriani et al., 2023). In contrast, incorrect thoughts collapse into flatter, less organized structures, reflecting a collapse of expressiveness and representational capacity (Ansuini et al., 2019; Cheng et al., 2025).

**Differences in thinking patterns become more distinguishable at later steps.** At the beginning of the thinking processes, the representation quality metrics change rapidly and show little difference between correct and incorrect latent thoughts. This probably reflects an exploratory reasoning phase, where the model is actively processing information and has not yet formed a clear solution path. As the thinking progresses, these metrics then stabilize and the difference between correct and incorrect thoughts becomes more salient, suggesting that the thinking process has converged to a solution, with the emergence of distinct reasoning patterns between correct and incorrect latent thoughts.

### 3.3 LATENT THOUGHTS ENCODE SIGNALS PREDICTIVE OF THEIR CORRECTNESS

Empirical results from Section 3.1 and Section 3.2 demonstrate that the latent thoughts contain rich semantic and geometric features that are predictive of their correctness. If these signals indeed capture the distinction between correct and incorrect thinking processes, they should be discriminative enough for a model to identify their correctness directly from the latent thoughts. To verify this hy-

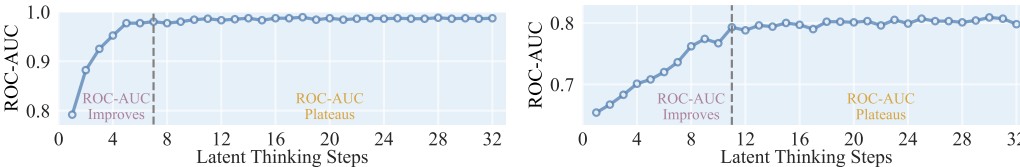

(a) ROC-AUC of the latent classifier on SVAMP.  (b) ROC-AUC of the latent classifier on MBPP.

Figure 3: Performance of the latent classifier trained with varying numbers of latent thinking steps on the SVAMP and MBPP datasets. Additional metrics and results are available in Appendix H.1

pothesis, we follow the widely-used probing technique (Liu et al., 2019; Hewitt & Manning, 2019), and train a lightweight sequence classifier to predict the correctness of latent thoughts.

The classifier takes as input the trajectory of latent thoughts from a problem, and predicts the probability that the thinking process is correct. For each problem in the training set, we sample 5 different latent thoughts and answers, and train the classifier to predict the correctness of the answer via binary cross-entropy loss. More training details of the latent classifier are in Appendix E. To study how predictive the latent representations are at different thinking steps, we construct 32 experimental settings for each dataset. In the $t$-th experiment ($1 \leq t \leq 32$), the classifier receives the first $t$ steps of latent thoughts $\mathbf{h}_{1:t}$ as input. The maximum of 32 steps is chosen to match the default number of thinking steps in Huginn-3.5B. Evaluation is performed on the test set using standard binary classification metrics such as ROC-AUC and Accuracy.

The results are shown in Figure 3. We observe that this latent classifier achieves strong performance on the test set, although it is trained with only a relatively small amount of data. On SVAMP, it achieves an ROC-AUC score close to 1.0, while on MBPP it achieves an ROC-AUC score of around 0.8. These results indicate that latent thoughts encode rich signals that are highly predictive of their correctness. We also observe that classification performance improves steadily with more thinking steps included, before reaching a plateau. This is consistent with the observation in Section 3.2 that differences between correct and incorrect thinking patterns become more distinguishable after a few thinking steps. Furthermore, the fact that incorporating latent thoughts from multiple steps improves classification performance suggests that the correctness signal is not solely reflected in a specific step of thought, but also in the evolving dynamics of the whole latent thinking trajectory.

> **Major Observation:** The latent reasoning language model displays distinct thinking patterns between correct and incorrect thinking processes, and such difference is highly distinguishable in the latent space, especially after a few thinking steps.

## 4 LATENT THINKING OPTIMIZATION

Motivated by our observations, we propose Latent Thinking Optimization (LTO), a probabilistic optimization approach designed to improve the latent thinking processes by selectively sampling trajectories that exhibit correct patterns. LTO formulates this as an optimization problem over latent policies, and introduces a probabilistic algorithm to solve the optimization problem. While LTO uses Huginn-3.5B as a starting point, we further demonstrate that LTO can also be applied to general LLMs, and achieves strong transferability across diverse domains with high efficiency. These results highlight the potential of LTO as an effective and scalable approach for optimizing LLM thinking by performing reward modeling and thinking correction directly in the latent space.

**Objective of LTO**  To formalize this, we introduce a binary variable $\mathcal{O}$ that indicates whether the latent thinking trajectory $z$ is correct. Our goal is to find an optimal latent thinking policy $\pi^*(z|x)$ such that it maximizes the expectation of generating a correct latent thinking trajectory $z$:

$$\pi^*(z \mid x) = \arg\max_{\pi(z|x)} \mathbb{E}_{z \sim \pi(z|x)} p(\mathcal{O} = 1 \mid x, z) \tag{1}$$

where $p(\mathcal{O} = 1|x, z)$ is the probability of a latent thinking trajectory $z$ being correct for a question $x$. Since the classifier introduced in Section 3.3 is trained to predict the correctness of latent thoughts,

---

**Algorithm 1** Latent Thinking Optimization

---

1: **Input:** question $x$, the original policy $\pi_{\text{ref}}(z|x)$, LRM $r(x, z)$, sampling budget $N$, the number of required samples $M$, weight $\beta > 0$ to control the strength of KL-regularization
2: **Output:** sampled set of latent thinking trajectories $\mathcal{C}$
3: $\mathcal{C} \leftarrow \emptyset, r_{\max} \leftarrow 0$  $\qquad\qquad\qquad\qquad\qquad\qquad$ ▷ Initialize the output set and the maximum reward.
4: **for** $i = 1$ **to** $N$ **do**
5: $\quad z_i \leftarrow z \sim \pi_{\text{ref}}(z|x)$  $\qquad\qquad$ ▷ Sample the $i$-th latent thinking trajectory from the original policy.
6: $\quad r_{\max} \leftarrow \max\{r_{\max}, r(z_i, x)\}$  $\qquad\qquad\qquad\qquad\qquad\qquad$ ▷ Update the maximum reward.
7: **while** $|\mathcal{C}| < M$ **do**  $\qquad\qquad\qquad\qquad\qquad\qquad$ ▷ Repeat until $M$ samples are collected.
8: $\quad z_i \sim \text{Uniform}\{z_j\}_{j=1}^N, u_i \sim \text{Uniform}(0, 1)$
9: $\quad \phi_i \leftarrow \exp((r(z_i, x) - r_{\max})/\beta)$  $\qquad\qquad\qquad$ ▷ Calculate the acceptance probability $\phi_i$.
10: $\quad$ **if** $u_i \geq \phi_i$ **then continue**  $\qquad\qquad\qquad$ ▷ Reject the sample $z_i$ with probability $1 - \phi_i$.
11: $\quad \mathcal{C} \leftarrow \mathcal{C} \cup \{z_i\}$  $\qquad\qquad\qquad$ ▷ Otherwise, accept the sample $z_i$ with probability $\phi_i$.
12: **return** $\mathcal{C}$

---

it can be used as a Latent Reward Model (LRM) $r(x, z)$ to estimate the probability of $p(\mathcal{O} = 1|x, z)$. To ensure that the optimized policy does not deviate significantly from the original policy $\pi_{\text{ref}}(z|x)$ (the latent policy distribution of the model before LTO optimization), we introduce a KL-regularization term (Jaques et al., 2017; Ziegler et al., 2019; Rafailov et al., 2023) with the weight $\beta$ to penalize the difference between the optimized policy $\pi^*(z|x)$ and the original policy $\pi_{\text{ref}}(z|x)$. The optimization objective then becomes:

$$\pi^*(z|x) = \arg\max_{\pi(z|x)} \mathbb{E}_{z \sim \pi(z|x)} [r(x, z)] - \beta \mathbb{D}_{\text{KL}}(\pi(z|x) || \pi_{\text{ref}}(z|x)) \qquad (2)$$

**The Necessity of KL regularization**  The inclusion of KL regularization with the weight $\beta$ is well motivated from the KL-regularized policy optimization theory. Without KL regularization, the optimized policy might collapse into a degenerated policy that significantly deviates from the base policy. Moreover, if we remove $\beta$ (set $\beta \to 0$), LTO will reduce to best-of-N sampling in the latent space, i.e., we sample N latent trajectories, score them with the LRM, and pick the highest-scoring one. This strategy only works well if the LRM is nearly perfect with almost 1.0 classification accuracy for correct and incorrect latent trajectories. In practice, the LRM is learned and not perfect. Without any regularization, LTO may exploit the errors of the LRM and select suboptimal latent thoughts. By including a KL penalty, we constrain the optimized policy so that it does not drift too far from the base policy, thereby preserving the diversity of sampled latent thoughts and mitigating overfitting to reward model noise.

**Probabilistic Sampling**  Directly optimizing over the latent policy $\pi(z|x)$ is often difficult. Instead, we approximate $\pi(z|x)$ using a finite set of $N$ sampled latent thinking trajectories $\{z_i\}_{i=1}^N$. In this case, we show that Equation 2 has a closed-form solution:

**Theorem 1.** *Given a sampled set of $\{z_i\}_{i=1}^N$ to approximate the policy distribution $\pi^*(z|x)$, for each $i$, the solution to Equation 2 is $\pi_r(z_i|x) = \frac{\pi_{ref}(z_i|x) \exp\left(\frac{1}{\beta} r(x, z_i)\right)}{\sum_{j=1}^N \pi_{ref}(z_j|x) \exp\left(\frac{1}{\beta} r(x, z_j)\right)}$.*

We provide the proof in Appendix F.1. Here we use the subscript notation $\pi_r$ to indicate that the policy is derived from the reward function $r(x, z)$. For simplicity, we omit the superscript $*$, but $\pi_r$ still represents the optimized policy.

While Theorem 1 gives a closed-form solution for our optimization problem, sampling directly from the distribution $\pi_r(z|x)$ is still difficult, since it is hard to accurately estimate the probability of each $\pi_{\text{ref}}(z_i|x)$. To address this issue, inspired by acceptance-rejection sampling algorithms (Flury, 1990; Grover et al., 2018; Liu et al., 2024), we propose Algorithm 1 to draw samples $z$ without explicitly calculating the value of $\pi_r(z|x)$. It draws $N$ candidate thinking trajectories $\{z_i\}_{i=1}^N$ from the original policy $\pi_{\text{ref}}(z|x)$. Each candidate sample $z_i$ will only be accepted with probability $\phi_i$, where $\phi_i$ is designed such that the latent thinking trajectories with higher reward are more likely to be accepted. This procedure is repeated until $M$ valid samples are collected. Theoretically, the set of samples drawn in Algorithm 1 is guaranteed to follow the distribution $\pi_r(z|x)$:

**Theorem 2.** *In Algorithm 1, for each $i$, the probability of $z_i$ being drawn and accepted is $\Pr(z_i|u_i < \phi_i, x) = \pi_r(z_i|x)$.*

We provide the proof in Appendix F.2. This theorem shows that each accepted sample $z_i$ is drawn with probability $\pi_r(z_i|x)$. Since each sampling process is independent, repeating the procedure produces i.i.d. samples from exactly the same distribution $\pi_r(z|x)$.

We summarize the workflow of LTO as follows: 1) Collect latent thinking trajectories from the training data to train LRM and 2) sample multiple latent thinking trajectories and accept only high-rewarded ones that are more likely to be correct via Algorithm 1. The samples drawn from Algorithm 1 is theoretically guaranteed to follow the distribution in Theorem 1, which is the solution to the objective of latent thinking optimization problem as defined in Equation 2. Note that although LTO relies on probabilistic sampling instead of parameter update to improve the latent policy, this does not diminish its nature as solving a discrete optimization problem over the latent policy.

**Application to General LLMs**  While our main focus is to improve the thinking process of the latent reasoning language model, the proposed LTO algorithm can also be applied to general LLMs, such as OLMo (Groeneveld et al., 2024), Llama (Touvron et al., 2023b) and Mistral (Jiang et al., 2023). Although general LLMs do not explicitly introduce a latent thinking process, their latent representations across multiple layers can be interpreted as latent chain of thoughts (Wang et al., 2025c). Under this view, LRM and LTO can be readily applied to general LLMs. In Appendix H.1, we demonstrate that LRMs trained with the latent representations of general LLMs can also achieve strong classification performance, indicating that the latent thoughts from general LLMs also encode appropriate reward signals. In Section 6.2, we demonstrate that LTO can significantly improve the performance of general LLMs on diverse reasoning tasks using these LRMs.

**Generalist Reward Modeling**  Natural language-based process reward models are often limited to narrow domains such as math (Wang et al., 2024; Lu et al., 2024) due to their reliance on domain-specific thinking formats and structures (Zeng et al., 2025). By comparison, reward modeling in the latent space has the potential for better generalizability, since latent thoughts share a unified form of latent representations and may be more transferable across diverse domains. In Section 6.3, we demonstrate that LRM achieves strong transferability across diverse domains and shows potential for building a generalist reward model in the latent space.

**High Efficiency**  LRM only requires a modest number of training samples (Section G.2), and LTO is highly efficient at both the training and inference stage (Section H.7). This highlights the potential of LTO as an efficient alternative that performs reward modeling in the latent space, in contrast to natural language-based reward models that require substantial finetuning and inference costs (Wang et al., 2024; Lu et al., 2024; Lightman et al., 2024).

**Guaranteed Performance Improvement**  While LTO does not explicitly modify the latent policy, we theoretically demonstrate in Appendix F.3 that improving the accuracy of the LRM directly translates into a higher expected correctness rate. Thus, LTO enables latent thinking improvement simply by scaling and refining the LRM (e.g., with more training data) which is computationally lightweight, rather than costly finetuning the base model to improve its latent policy.

## 5  EXPERIMENTAL SETTINGS

**Datasets**  To study whether our approach can improve the thinking processes of Huginn-3.5B across diverse tasks with different thinking patterns, we evaluate its performance on five datasets from three domains: (1) **GSM8K** (Cobbe et al., 2021), **SVAMP** (Patel et al., 2021), **GSM-Symbolic** (Mirzadeh et al., 2025) for the *Math* domain, (2) **CommonsenseQA** (Talmor et al., 2019) for the *Commonsense Reasoning* domain; and (3) **MBPP** (Austin et al., 2021) for the *Code Generation* domain. The details of the datasets are in Appendix G.1.

**Baselines and Implementation Details**  Since Huginn-3.5B generates the thinking process in the form of latent representations, many thinking correction methods with a trained process verifier in the natural language space (Lu et al., 2024; Wang et al., 2024) may not be applicable to the latent space. Therefore, we compare our approach against two types of reasoning correction and improvement methods applicable to Huginn-3.5B: (1) *Answer Correction*. These methods correct and improve the answers without requiring access to the thinking process. We include three

Table 1: Comparison of the answer correctness rate of Huginn-3.5B using different correction methods. The best performance in each column is in **bold**, and the performance of the best baseline in each column is underlined. $*$ indicates statistically significant improvement with $p < 0.05$.

| Method | GSM8K | GSM-Symbolic | SVAMP | CommonsenseQA | MBPP |
|---|---|---|---|---|---|
| Base Model | 0.326 | 0.265 | 0.517 | 0.500 | 0.278 |
| Majority Voting | 0.333 | 0.269 | 0.511 | 0.504 | 0.288 |
| Self-Correction w. Confidence Score | 0.342 | 0.281 | 0.524 | 0.507 | 0.288 |
| Self-Correction w. Verbal Evaluation | 0.262 | 0.193 | 0.518 | 0.505 | 0.226 |
| Latent Thinking Correction w. CoE-R | 0.330 | 0.259 | 0.510 | 0.504 | 0.276 |
| Latent Thinking Correction w. CoE-C | 0.324 | 0.256 | 0.516 | 0.507 | 0.280 |
| Weighted Majority Voting w. LRM | 0.375$*$ | 0.301$*$ | 0.537$*$ | 0.509 | 0.295$*$ |
| Latent Thinking Optimization w. LRM | **0.385**$*$ | **0.305**$*$ | **0.538**$*$ | **0.517**$*$ | **0.299**$*$ |

representative approaches: Majority Voting (Wang et al., 2023), Self-Correction with Confidence Score (Ren et al., 2023b), Self-Correction with Verbal Evaluation (Manakul et al., 2023). (2) *Latent Thinking Correction*. While explicit correction of latent thoughts remains underexplored, a recent work (Wang et al., 2025c) introduces two heuristic metrics (CoE-R and CoE-C) to evaluate the correctness score of latent thoughts. We adopt these scores as the correction signals, yielding two additional baselines: Latent Thinking Correction with CoE-R, and Latent Thinking Correction with CoE-C. Furthermore, we evaluate a simplified version of our approach, Weighted Majority Voting with LRM, which use the LRM reward as a weighting signal. We also report the performance of the base model (directly generating a latent thinking trajectory and the corresponding answer without any correction) to quantify the performance improvement achieved by our approach and competing baselines. Implementation details of baselines and our method are in Section G.2.

# 6 Experimental Results

## 6.1 Overall Performance Comparison

Table 1 presents the experimental results on all the datasets. We have the following observations:

**LTO significantly improves the latent thinking processes.** Across all datasets, LTO consistently outperforms both the base model and the best baseline for thinking correction. Leveraging a well-trained LRM, LTO can effectively detect and correct erroneous thinking patterns in the latent space via a probabilistic algorithm, bringing robust and consistent improvements to the latent thinking processes. By comparison, other thinking correction methods show suboptimal performance or even worse performance than the base model, indicating that these techniques originally developed for verbal thinking are not suitable for identifying errors for latent thinking.

**LRM is highly effective in detecting incorrect latent thinking patterns.** Both weighted majority voting and Latent Thinking Optimization with LRM achieve consistent improvements over the base model. Notably, standard majority voting yields little to no benefit; however, when the LRM reward is used as a weighting signal, weighted majority voting achieves substantial gains. This demonstrates that the LRM reward provides a reliable estimation of the correctness of latent thoughts and serves as an effective correction signal for thinking correction algorithms in the latent space.

## 6.2 Application to General LLMs

While we mainly focus on improving the thinking process of Huginn-3.5B, we also demonstrate that LTO can be applied to general LLMs, such as OLMo (Groeneveld et al., 2024), Llama (Touvron et al., 2023b) and Mistral (Jiang et al., 2023). To evaluate the performance of LTO on general LLMs, we use the same LRM and LTO configurations as described in Section 5, and train LRMs using the latent representations from general LLMs. From the experimental results in Table 2, we can see that LTO achieves substantial performance gains across diverse datasets, with an improvement of up to 103% over the base model, even with a modest sampling budget ($N = 20$). These results highlight the potential of LTO as a general method for improving the latent thinking processes of LLMs.

Table 2: Performance of LTO on general LLMs. The best-performing method for each model is in **bold**. ∗ indicates the improvement over the best runner-up is statistically significant with $p < 0.05$.

| Model | Method | GSM8K | GSM-Symbolic | SVAMP | CommonsenseQA | MBPP |
|---|---|---|---|---|---|---|
| OLMo-7B | Base Model | 0.124 | 0.078 | 0.297 | 0.464 | 0.244 |
| | Majority Voting | 0.209 | 0.149 | 0.469 | 0.521 | 0.240 |
| | Latent Thinking Optimization | **0.252**∗ | **0.154**∗ | **0.552**∗ | **0.602**∗ | **0.308**∗ |
| Llama-2-7B | Base Model | 0.223 | 0.204 | 0.473 | 0.399 | 0.189 |
| | Majority Voting | 0.275 | 0.302 | 0.598 | 0.493 | 0.193 |
| | Latent Thinking Optimization | **0.389**∗ | **0.316**∗ | **0.776**∗ | **0.606**∗ | **0.237**∗ |
| Llama-2-13B | Base Model | 0.306 | 0.273 | 0.521 | 0.398 | 0.247 |
| | Majority Voting | 0.417 | 0.379 | 0.612 | 0.501 | 0.263 |
| | Latent Thinking Optimization | **0.534**∗ | **0.442**∗ | **0.791**∗ | **0.650**∗ | **0.322**∗ |
| Mistral-7B | Base Model | 0.368 | 0.278 | 0.548 | 0.671 | 0.315 |
| | Majority Voting | 0.529 | 0.413 | 0.624 | 0.687 | 0.334 |
| | Latent Thinking Optimization | **0.565**∗ | **0.462**∗ | **0.771**∗ | **0.708**∗ | **0.388**∗ |

## 6.3 GENERALIST REWARD MODELING

To evaluate whether LRMs trained on one dataset can be applied to another dataset, we first examine the cross-dataset transferability of LRMs by evaluating the performance of LTO when paired with an LRM trained on different datasets. We extend the study by training a general LRM on the combined training data from all datasets and evaluating the performance of LTO with the general LRM. From the results in Figure 4, we can see that LRMs demonstrate transferability across different domains, since LTO can improve the performance of the base model when paired with an LRM trained on a different dataset. The improvement is consistent even if the gap between domains is large. For example, although CommonsenseQA primarily involves commonsense reasoning, an LRM trained on CommonsenseQA still improves performance on math-focused datasets such as GSM8K, GSM-Symbolic, and SVAMP. This suggests that LRMs may capture some fundamental aspects of latent thinking patterns shareable across different domains. Furthermore, the performance of LTO using the general LRM is on par with the performance of LTO using domain-specific LRMs.

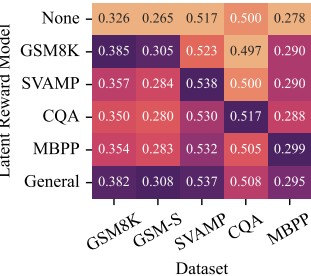

Figure 4: Performance of LTO using different LRMs. "GSM-S" refers to the GSM-Symbolic dataset. "CQA" refers to the CommonsenseQA dataset. "None" refers to the performance of the base model without LTO.

These results suggest that latent reward modeling can generalize across domains. While our empirical results have not yet achieved full transferability across all possible tasks, we believe that they indicate promising cross-domain potential for building a generalist reward model in the latent space for future work.

## 7 CONCLUSION

In this paper, we observe that the latent thoughts of Huginn-3.5B that lead to correct versus incorrect answers display distinct thinking patterns, and such difference is highly distinguishable by a latent classifier. Building on these insights, we formulate latent thinking improvement as a reward optimization problem over latent policies, and propose an LTO algorithm that uses the latent classifier as an LRM to optimize the latent thinking processes. Extensive experiments across diverse reasoning tasks demonstrate LTO can significantly improve the latent thinking processes of Huginn-3.5B. Furthermore, we show LRM can generalize across different domains, and LTO can be seamlessly applied to general LLMs to improve their thinking processes. In contrast to verbal thinking approaches that scale test-time compute through natural language generation (Guo et al., 2025; Muennighoff et al., 2025), our method demonstrates that reward modeling and scaling test-time thinking with supervision can be performed directly in the latent space, offering a general (Section 6.2), efficient (Appendix H.7), and domain-agnostic (Section 6.3) approach to improving the thinking processes of LLMs. We discuss the related works, limitations, broader impact and reproducibility of our research in Appendix A, Appendix I, Appendix J and Appendix L, respectively.

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

# Appendices

## A RELATED WORKS

### A.1 VERBAL AND LATENT THINKING FOR LLMS

Human cognition often involves thinking through intermediate steps rather than directly answering the question (Kahneman, 2011; Zelikman et al., 2024). Inspired by this, a growing line of research focuses on guiding LLMs to generate intermediate reasoning steps as the thinking process before generating the answers. Most approaches represent the thinking process in natural language, such as step-by-step chain-of-thought prompting (Wei et al., 2022; Kojima et al., 2022; Wang et al., 2023), self-correction via iterative feedback (Shinn et al., 2023; Madaan et al., 2023; Kumar et al., 2025), or building reasoning trees to explore diverse solutions (Yao et al., 2023; Hao et al., 2023). While effective, such verbal thinking incurs significant computational cost, and is also prone to the over-thinking issue (Chen et al., 2025; Sui et al., 2025). In contrast, latent thinking offers an alternative approach, where the model represents its thinking process as compact latent representations rather than natural language. This approach is more computationally efficient and better suited for reasoning with abstract concepts that are difficult to verbalize. Among various approaches for latent thinking (Zhang et al., 2023; Goyal et al., 2024; Hao et al., 2025; Geiping et al., 2025), a representative one is the latent reasoning language model (Geiping et al., 2025), which is pretrained from scratch as a new language model architecture. It introduces a recurrent unit to generate sequences of latent thoughts and supports test-time scaling with flexible computation budgets. Despite promising, the lack of interpretability in the latent representations makes it difficult to understand what the model is actually thinking about or to verify the correctness of its thinking process. *In this paper, we aim to bridge this gap by investigating how the latent reasoning language model thinks in the latent space and how external supervision can guide and improve the latent thinking processes.*

### A.2 SCALING UP TEST-TIME COMPUTE

As LLMs are tasked with increasingly difficult problems, directly prompting the LLM to generate the answers is often insufficient. To address this, recent works emphasize scaling up test-time compute as an effective approach to enhance the problem-solving capability of LLMs (Sardana et al., 2024; Snell et al., 2025). Existing approaches scale up test time compute from different perspectives, such as sequential scaling with revisions to refine the answer (Shinn et al., 2023; Madaan et al., 2023; Muennighoff et al., 2025), parallel scaling by generating multiple answers to search for diverse solutions (Wang et al., 2023; Yao et al., 2023; Hao et al., 2023), or scaling with a verifier or reward model to ensure the correctness of solutions (Wang et al., 2024; Lu et al., 2024; Feng et al., 2025; Setlur et al., 2025). However, most of these approaches focus on scaling up test-time compute using natural language, and how to scale up test-time compute in the latent space (Geiping et al., 2025) remains underexplored. *In this paper, we introduce a probabilistic sampling approach with a latent reward model that can improve the latent thinking processes and enable efficient and effective test-time scaling in the latent space.*

## B ADDITIONAL VISUALIZATION OF LATENT THOUGHTS

Additional examples on the visualization of correct and incorrect latent thoughts are in Figure A1.

## C CALCULATION OF REPRESENTATION QUALITY METRICS

In this section, we provide the details on how to calculate the representation quality metrics. For a question $x$, Huginn-3.5B generates $T$ steps of latent thoughts $\mathbf{h}_{1:T}$ recursively. Each latent thought $\mathbf{h}_t \in \mathbb{R}^{L \times d}$ is an internal hidden state generated by Huginn-3.5B, where $L$ is the number of tokens, $d$ is the hidden dimensionality. The representation quality metrics are calculated over the latent thoughts across all the thinking steps to capture the evolving dynamics of latent thinking processes. Note that in the Huginn-3.5B architecture, the same RMSNorm module with the same rescaling weight $w$ is applied to each step of latent thought. Therefore, all the latent thoughts from different

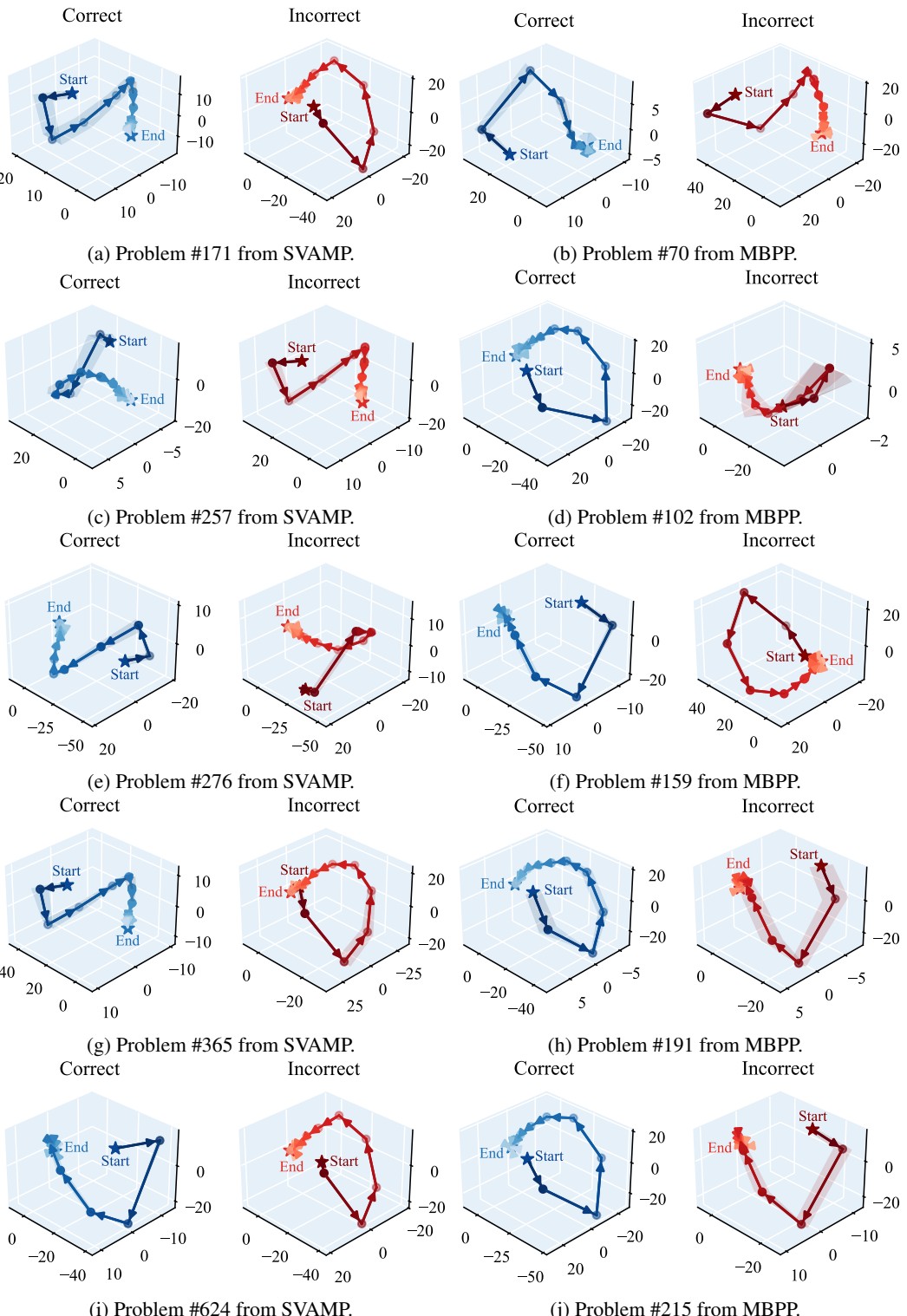

Figure A1: Visualization of the distribution of the correct and incorrect latent thoughts projected onto 3D space demonstrate that correct and incorrect latent thoughts exhibit different patterns in the latent space. Note that this phenomenon is not limited to these cases. On the SVAMP dataset, we identify 1,654 problems with both correct and incorrect answers, and on the MBPP dataset, we identify 179 problems with both correct and incorrect answers. In all of these cases, the latent thoughts leading to correct versus incorrect answers show different patterns in the latent space.

steps are already normalized to the same scale before we calculate the representation quality metrics, making these latent representations scale-invariant. For each latent thought $\mathbf{h}_t (1 \leq t \leq T)$, we calculate the Entropy, Effective Rank, Anisotropy and Intrinsic Dimension of $\mathbf{h}_t$ as follows.

## C.1 ENTROPY

Entropy (Skean et al., 2025) quantifies how much information content the latent representations carry. A higher entropy indicates a richer spread of information across many dimensions, reflecting diverse, less redundant features and better information preservation. Conversely, a lower entropy reflects concentrated eigenvalue spectra, suggesting that the latent representations may contain redundant information. We compute the entropy over the Gram matrix $\mathbf{K} = \mathbf{h}_t \mathbf{h}_t^\top$ using the matrix-based Rényi entropy. For any $\alpha > 0$, this is defined as:

$$\text{Entropy}(\mathbf{h}_t) = \frac{1}{1-\alpha} \log \left( \sum_{i=1}^{r} \left( \frac{\lambda_i(\mathbf{K})}{\text{tr}(\mathbf{K})} \right)^\alpha \right), \tag{3}$$

where $\lambda_i(\mathbf{K})$ denotes the $i$-th eigenvalue of the Gram matrix $\mathbf{K}$, $r = \text{rank}(\mathbf{K})$ denotes its rank. While we can vary $\alpha$ to get different formulations of matrix entropy, we follow the approach of Skean et al. (2025) and choose $\alpha \to 1$, which is equivalent to the standard von Neumann entropy.

## C.2 EFFECTIVE RANK

Effective Rank (Wei et al., 2024) measures how effectively the model extracts key concepts and reduces noisy features in its latent representations. A higher effective rank implies that the representations contain noisy features, while a lower effective rank indicates better noise reduction. It is defined as follows:

$$\text{EffectiveRank}(\mathbf{h}_t) = \exp \left( -\sum_{i=1}^{K} \frac{\sigma_i}{\sum_{i=1}^{K} \sigma_i} \log \frac{\sigma_i}{\sum_{i=1}^{K} \sigma_i} \right), \tag{4}$$

where $K = \min\{L, d\}$, and $\sigma_1, \sigma_2, \ldots, \sigma_K$ are the singular values of the matrix $\mathbf{h}_t$.

## C.3 ANISOTROPY

Anisotropy (Razzhigaev et al., 2024) measures the non-uniformity of a distribution in the latent space. A higher anisotropy suggests that representations are more directed in specific orientations, while a lower anisotropy indicates that the representations are spread out more evenly in all directions. It is defined as follows:

$$\text{Anisotropy}(\mathbf{h}_t) = \frac{\sigma_1^2}{\sum_{i=1}^{K} \sigma_i^2}. \tag{5}$$

where $K = \min\{L, d\}$, and $\sigma_1, \sigma_2, \ldots, \sigma_K$ are the singular values of the matrix $\mathbf{h}_t$.

## C.4 INTRINSIC DIMENSION

Intrinsic Dimension (Facco et al., 2017; Cheng et al., 2025) quantifies the minimal number of coordinates required to describe the local geometric structure of the representations without significant information loss. A higher intrinsic dimension indicates a rich, complex latent structure, while a lower intrinsic dimension suggests the representation lies on a simpler manifold. Specifically, for the matrix $\mathbf{h}_t \in \mathbb{R}^{L \times d}$, we can view it as a collection of $L$ points $\mathbf{h}_t^i$ in the $d$-dimensional space, i.e., $\mathbf{h}_t = \{\mathbf{h}_t^i\}_{i=1}^L$. To calculate the intrinsic dimension, we use the Two-Nearest-Neighbour estimator (Facco et al., 2017): for each point $\mathbf{h}_t^i$, we compute its nearest-neighbor distance $r_{1,i}$ and second-nearest-neighbor distance $r_{2,i}$, and form the ratio $\mu_i = r_{2,i}/r_{1,i}$. Sorting $\{\mu_i\}_{i=1}^L$ in ascending order yields $\mu_{(1)}, \ldots, \mu_{(L)}$, and the empirical cumulative distribution is given by $F_j = j/L$. Each $\mu_{(j)}$ is then mapped to a transformed data point ($x_j = \log \mu_{(j)}, y_j = -\log(1 - F_j)$). Under mild assumptions, the points $\{(x_j, y_j)\}_{j=1}^L$ are theoretically expected to align on a straight line through the origin, and the slope of this line provides an estimation of the intrinsic dimension. Following Glielmo et al. (2022), we use the standard Euclidean distance as the distance metric, and

introduce a trimming factor $f = 0.1$ to discard extremely large values of $\mu_i = r_{2,i}/r_{1,i}$, ensuring robustness against outlier data points that may violate the estimator's assumptions. The detailed algorithm is summarized in Algorithm 2.

---

**Algorithm 2** Calculation of Intrinsic Dimension with Two-Nearest-Neighbour Estimation

1: **Input:** matrix $\mathbf{h}_t = \{\mathbf{h}_t^i\}_{i=1}^L$, distance metric $\text{dist}(\cdot, \cdot)$, trimming fraction $f \in [0, 1)$
2: **Output:** estimated intrinsic dimension $\hat{d}$
3: **for** $i = 1$ **to** $L$ **do**
4:      Compute the pairwise distances $\{\text{dist}(\mathbf{h}_t^i, \mathbf{h}_t^j)\}_{j \neq i}$
5:      $r_{1,i} \leftarrow$ smallest distance (nearest neighbor)
6:      $r_{2,i} \leftarrow$ second smallest distance (second nearest neighbor)
7:      $\mu_i \leftarrow r_{2,i}/r_{1,i}$
8: Sort $\{\mu_i\}_{i=1}^L$ in ascending order to obtain $\mu_{(1)}, \ldots, \mu_{(L)}$
9: **for** $j = 1$ **to** $L$ **do**
10:      $F_j \leftarrow j/L$
11:      $x_j \leftarrow \log(\mu_{(j)})$
12:      $y_j \leftarrow -\log(1 - F_j)$
13: **if** $f > 0$ **then**
14:      Trim the largest $\lceil f \cdot L \rceil$ values of $\mu_{(j)}$ by setting $L' \leftarrow \lfloor (1 - f) L \rfloor$
15: **else**
16:      Keep all the $\mu_{(j)}$ by setting $L' \leftarrow L$
17: Fit the points of the plane given by coordinates $\{(x_j, y_j)\}_{j=1}^{L'}$ with a straight line $y = \hat{d} \cdot x$ passing through the origin
18: **return** the slope $\hat{d}$ as the estimated intrinsic dimension

---

# D    INTERPRETABILITY ANALYSIS OF LATENT THOUGHTS

To better understand the observed patterns in the correct and incorrect latent thoughts, we provide an interpretability analysis by decoding the latent thoughts at different reasoning steps and examining how they evolve toward (or away from) the correct answer. To make this analysis clear, we use a one-digit arithmetic dataset, where the model must output a single digit answer to an arithmetic question. This setting is ideal for interpretability because the operations are simple and easy to verify, and we can directly inspect how the decoded latent thoughts evolve at the specific token position corresponding to the answer digit. Using the coda module (decoder) from the latent reasoning language model, we decode the latent thought at each thinking step into its top-5 most probable tokens and analyze the progression of latent thoughts and show representative examples with correct and incorrect thinking patterns in Figure A2.

For correct examples, we observe that the correct digit token emerges among the top-k candidates at middle thinking steps, then rises to rank-1 and stabilizes in later steps. In contrast, for incorrect examples, the correct token does not consistently rise in rank, the latent thoughts show fluctuation or drifting behavior and the final step fails to converge to the correct answer. The pattern differences between correct and incorrect latent thoughts demonstrate that the latent reasoning language model encodes meaningful reasoning patterns in its latent thoughts that reflect its thinking processes.

# E    TRAINING DETAILS OF THE LATENT CLASSIFIER

To capture the thinking dynamics of the latent thoughts across different thinking steps, we design a latent classifier that can operate over the sequence of latent representations. Specifically, we adopt a 2-layer Transformer (Vaswani et al., 2017) with Sinusoidal positional encoding to encode the sequence of latent thoughts. The configuration of the latent classifier (hidden dimensionality 5280, number of attention heads 55, and MLP hidden size 17920) follows the configuration of Huginn-3.5B. While we observe in our experiments that alternative configurations also bring comparable performance, we use this configuration as the default setting. The output sequences of the Transformer are aggregated with mean pooling over the dimension $T$ (number of thinking steps), followed by a two-layer MLP with ReLU as activation function to produce logits for binary

**Example 1 with Correct Thinking Patterns (Answer: 7)**

Question: What is (1 * 1) + 6? Answer:
Top-5 ranked tokens decoded from latent thoughts at different
thinking steps:
Step 16: [-, 1, 3, 2, 4]
Step 32: [1, 7, 6, 2, 8]
Step 48: [7, 6, 1, 8, 2]
Step 64: [7, 6, 1, 8, 2]

----

**Example 2 with Correct Thinking Patterns (Answer: 5)**

Question: What is (7 + 2) - 4? Answer:
Top-5 ranked tokens decoded from latent thoughts at different
thinking steps:
Step 16: [-, 1, 3, 2, 4]
Step 32: [1, 5, 9, 2, 6]
Step 48: [5, 1, -, 7, 4]
Step 64: [5, 7, 9, 1, -]

----

**Example 3 with Correct Thinking Patterns (Answer: 7)**

Question: What is (3 * 2) + 1? Answer:
Step 16: [-, 1, 3, 2, 4]
Step 32: [6, 1, 5, 7, 4]
Step 48: [6, 1, 7, 5, 9]
Step 64: [7, 1, 6, 5, 9]

----

**Example 4 with Incorrect Thinking Patterns (Answer: 8)**

Question: What is (8 - 2) + 2? Answer:
Top-5 ranked tokens decoded from latent thoughts at different
thinking steps:
Step 16: [-, 1, 2, 3, 4]
Step 32: [6, 4, 1, -, 5]
Step 48: [6, 4, -, 5, 8]
Step 64: [6, 4, 8, -, 1]

----

**Example 5 with Incorrect Thinking Patterns (Answer: 7):**

Question: What is (3 - 2) + 6? Answer:
Top-5 ranked tokens decoded from latent thoughts at different
thinking steps:
Step 16: [-, 1, 3, 2, 4]
Step 32: [1, -, 5, 2, 3]
Step 48: [3, 4, 5, 2, 1]
Step 64: [4, 5, 6, 3, 2]

----

**Example 6 with Incorrect Thinking Patterns (Answer: 7):**

Question: What is (6 - 4) + 5? Answer:
Top-5 ranked tokens decoded from latent thoughts at different
thinking steps:
Step 16: [-, 1, 3, 2, 4]
Step 32: [1, -, 2, 4, 3]
Step 48: [2, 1, 3, 4, 6]
Step 64: [1, 2, 3, 5, 4]

Figure A2: Tokens decoded from correct and incorrect latent thoughts at different thinking steps.

Table A1: Performance comparison of the latent classifier with different aggregation strategies. The best performance in each column is in **bold**.

| Aggregation Strategy | GSM8K | | SVAMP | | CommonsenseQA | | MBPP | |
|---|---|---|---|---|---|---|---|---|
| | Accuracy | ROC-AUC | Accuracy | ROC-AUC | Accuracy | ROC-AUC | Accuracy | ROC-AUC |
| first 10 tokens | 0.708 | 0.742 | 0.924 | 0.980 | 0.574 | 0.595 | 0.732 | 0.742 |
| last 10 tokens | 0.790 | 0.863 | 0.957 | **0.987** | 0.610 | 0.644 | 0.749 | 0.773 |
| all the tokens | **0.820** | **0.884** | **0.960** | 0.987 | **0.623** | **0.671** | **0.790** | **0.807** |

classification. Training is performed with binary cross-entropy loss for 10 epochs using the Adam optimizer (Kingma & Ba, 2015) with a learning rate of $5e - 6$.

However, a challenge is that each latent thought $\mathbf{h}_t \in \mathbb{R}^{L \times d}$ is a matrix rather than a vector, and this requires an aggregation over the dimension $L$ before it can be processed by the Transformer. To address this challenge, we experiment with different aggregation strategies in Table A1, and empirically we observe that apply mean pooling over the hidden states corresponding to all the $L$ tokens yields better performance than mean pooling over the hidden states corresponding to the first 10 or the last 10 tokens. Therefore, we choose to apply mean pooling over the dimension $L$ for each latent thought $\mathbf{h}_t$. This design choice is also motivated by the common practices in probing methods, where mean pooling over the sequence dimension is widely adopted as a standard approach for deriving fixed-length representations from variable-length sequences (Hewitt & Manning, 2019; Tenney et al., 2019; Ren et al., 2023a).

## F  ADDITIONAL THEORETICAL RESULTS

### F.1  PROOF FOR THEOREM 1

**Theorem 1.** *Given a sampled set of $\{z_i\}_{i=1}^N$ to approximate the policy distribution $\pi^*(z|x)$, for each $z_i$, the solution to Equation 2 is $\pi_r(z_i|x) = \frac{\pi_{ref}(z_i|x)\exp\left(\frac{1}{\beta}r(x,z_i)\right)}{\sum_{j=1}^N \pi_{ref}(z_j|x)\exp\left(\frac{1}{\beta}r(x,z_j)\right)}$.*

*Proof.* Since we are sampling from a discrete set of $\{z_i\}_{i=1}^N$, we represent the policy distribution $\pi(z|x)$ as a vector over the set of latent thoughts $\{z_i\}_{i=1}^N$. To ensure that $\pi(z|x)$ forms a valid policy distribution, $\pi(z|x)$ should satisfy the constraint $\sum_{i=1}^N \pi(z_i|x) = 1$. To solve the optimization problem from Equation 2 subject to this constraint, we introduce a Lagrange multiplier $\lambda$ and construct the Lagrangian:

$$\mathcal{L}(\pi(z|x), \lambda) = \sum_{i=1}^N \left[ \pi(z_i|x)r(x, z_i) - \beta\pi(z_i|x)\log\frac{\pi(z_i|x)}{\pi_{\text{ref}}(z_i|x)} \right] + \lambda \sum_{i=1}^N (\pi(z_i|x) - 1)$$

To find the solution to this problem, since we are optimizing over a probability distribution $\pi(z|x)$, we can compute the partial derivative of the objective $\mathcal{L}(\pi(z|x), \lambda)$ with respect to each coordinate $\pi(z_i|x)$. Setting the partial derivative to zero, for each $z_i$, we have:

$$\frac{\partial \mathcal{L}(\pi(z|x), \lambda)}{\partial \pi(z_i|x)} = r(x, z_i) - \beta\left(\log\frac{\pi(z_i|x)}{\pi_{\text{ref}}(z_i|x)} + 1\right) + \lambda = 0$$

By rearranging this equation, we can get:

$$\frac{\pi(z_i|x)}{\pi_{\text{ref}}(z_i|x)} = \exp(\frac{r(x, z_i) + \lambda - \beta}{\beta}) \Rightarrow \pi(z_i|x) \propto \pi_{\text{ref}}(z_i|x)\exp(\frac{r(x, z_i)}{\beta})$$

Plugging in the constraint that $\sum_{i=1}^N \pi(z_i|x) = 1$, for each $z_i$, we obtain the solution:

$$\pi_r(z_i|x) = \frac{\pi_{\text{ref}}(z_i \mid x)\exp\left(\frac{1}{\beta}r(x, z_i)\right)}{\sum_{j=1}^N \pi_{\text{ref}}(z_j \mid x)\exp\left(\frac{1}{\beta}r(x, z_j)\right)}$$

Here we use the subscript notation $\pi_r$ to indicate that the policy is derived from the reward function $r(x, z)$. For simplicity, we omit the superscript $*$, but $\pi_r$ still represents the optimized policy.

Intuitively, the optimized policy $\pi_r$ reweights the original policy $\pi_{\text{ref}}$ with the exponential reward term $\exp(\frac{1}{\beta}r(x, z))$: latent thinking trajectories with higher reward $r(x, z)$ will have higher probability of being selected, while trajectories with lower reward will have lower probability of being selected. The weight $\beta$ controls how strong this adjustment is: when $\beta$ is small, the policy becomes more "greedy" and focuses heavily on the high-rewarded latent thinking trajectories; when $\beta$ is large, it stays closer to the original policy $\pi_{\text{ref}}$.

## F.2  PROOF FOR THEOREM 2

**Theorem 2.** *In Algorithm 1, for each $i$, the probability of $z_i$ being drawn and accepted is $\Pr(z_i | u_i < \phi_i, x) = \pi_r(z_i | x)$.*

*Proof.* In Algorithm 1, since the distribution $\pi_r(z|x)$ is difficult to directly sample from, we would like to draw candidate samples $z$ from the distribution $\pi_{\text{ref}}(z|x)$, and only accept those samples that follow the distribution $\pi_r(z|x)$ with probability $\frac{\pi_r(z|x)}{M \cdot \pi_{\text{ref}}(z|x)}$. Here $M$ is a constant, and for the acceptance probability to be valid, it must satisfy $\frac{\pi_r(z|x)}{M \cdot \pi_{\text{ref}}(z|x)} \leq 1$, that is, $M \geq \frac{\pi_r(z_i|x)}{\pi_{\text{ref}}(z_i|x)}$ for each $z_i$. We choose the smallest possible $M$ so that each $z_i$ has the highest chance of being accepted, because a tight $M$ avoids unnecessary rejections and makes the algorithm more efficient. Therefore, the value of $M$ can be calculated as:

$$M = \max_{1 \leq i \leq N} \left\{ \frac{\pi_r(z_i|x)}{\pi_{\text{ref}}(z_i|x)} \right\} = \max_{1 \leq i \leq N} \left\{ \frac{\exp\left(\frac{1}{\beta}r(x, z_i)\right)}{\sum_{j=1}^{N} \pi_{\text{ref}}(z_j \mid x) \exp\left(\frac{1}{\beta}r(x, z_j)\right)} \right\}$$

$$= \frac{\exp\left(\frac{1}{\beta}r_{\max}\right)}{\sum_{j=1}^{N} \pi_{\text{ref}}(z_j \mid x) \exp\left(\frac{1}{\beta}r(x, z_j)\right)}$$

where $r_{\max}$ is the maximum reward calculated in Algorithm 1. Then we can get the acceptance probability $\phi_i$ for each $z_i$:

$$\phi_i = \frac{\pi_r(z_i|x)}{M \cdot \pi_{\text{ref}}(z_i|x)} = \frac{\exp\left(\frac{1}{\beta}r(x, z_i)\right)}{M \cdot \sum_{j=1}^{N} \pi_{\text{ref}}(z_j \mid x) \exp\left(\frac{1}{\beta}r(x, z_j)\right)} = \exp((r(x, z_i) - r_{\max})/\beta)$$

For each candidate $z_i$ we have:

$$\Pr(z_i, \ u_i < \phi_i \mid x) = \pi_{\text{ref}}(z_i \mid x) \cdot \phi_i = \pi_{\text{ref}}(z_i \mid x) \cdot \frac{\pi_r(z_i|x)}{M \cdot \pi_{\text{ref}}(z_i|x)} = \frac{\pi_r(z_i|x)}{M}.$$

The total probability of acceptance is:

$$\Pr(u_i < \phi_i \mid x) = \sum_{j=1}^{N} \Pr(z_j, \ u_i < \phi_i \mid x) = \sum_{j=1}^{N} \frac{\pi_r(z_j|x)}{M} = \frac{1}{M} \sum_{j=1}^{N} \pi_r(z_j|x) = \frac{1}{M}.$$

Therefore, by Bayes rule, the probability of $z_i$ being drawn and accepted is:

$$\Pr(z_i \mid u_i < \phi_i, x) = \frac{\Pr(z_i, \ u_i < \phi_i, \mid x)}{\Pr(u_i < \phi_i \mid x)} = \frac{\frac{\pi_r(z_i|x)}{M}}{\frac{1}{M}} = \pi_r(z_i|x).$$

## F.3  THEORETICAL ANALYSIS ON CORRECTNESS RATE

To analyze the expected correctness rate of the LTO algorithm using the trained latent classifier as LRM, we first introduce the notion of a *perfect reward model*, which serves as an oracle for evaluating the correctness of latent thinking trajectories. This formalization provides a reference point for quantifying the performance of the latent policy derived from the trained LRM:

**Definition 1** (Perfect reward model). *A perfect reward model $r^*(x, z)$ is a function that always assigns a value of 1.0 if the latent thinking trajectory $z$ is correct for question $x$, and 0.0 if the latent thinking trajectory $z$ is incorrect for question $x$. Using this definition, for a question $x$, the expected correctness rate of a latent policy $\pi$ can be represented as $\mathbb{E}_{z \sim \pi} r^*(x, z)$.*

Next, we introduce the following theorem to measure how the expected correctness rate of $z \sim \pi_r(z|x)$ (the policy derived from the trained LRM) relates to that of $z \sim \pi_{r^*}(z|x)$ (the policy derived from the perfect reward model):

**Theorem 3.** *For a question $x$, for each sample $z_i$, if the error between the trained reward model $r(x, z_i)$ and the perfect reward model $r^*(x, z_i)$ is bounded by $\epsilon$, that is, $|r(x, z_i) - r^*(x, z_i)| \leq \epsilon$, then the performance gap of using an imperfect reward model is upper bounded by $\left| \mathbb{E}_{z \sim \pi_r(z|x)} r^*(x, z) - \mathbb{E}_{z \sim \pi_{r^*}(z|x)} r^*(x, z) \right| \leq \sqrt{\frac{4\epsilon}{\beta}}$*

*Proof.* The expectation of the performance gap $\Delta$ between using the trained reward model and using the perfect reward model is:

$$\Delta = \left| \mathbb{E}_{z \sim \pi_r(z|x)} r^*(x, z) - \mathbb{E}_{z \sim \pi_{r^*}(z|x)} r^*(x, z) \right|$$

$$= \sum_{i=1}^{N} |\pi_r(z_i|x) - \pi_{r^*}(z_i|x)| \cdot r^*(x, z_i)$$

$$\leq \sum_{i=1}^{N} |\pi_r(z_i|x) - \pi_{r^*}(z_i|x)| \cdot 1$$

Using Pinsker's inequality (Cover & Thomas, 2006), we have:

$$\sum_{i=1}^{N} |\pi_r(z_i|x) - \pi_{r^*}(z_i|x)| \leq \sqrt{2\mathbb{D}_{\mathrm{KL}}(\pi_r(z|x)||\pi_{r^*}(z|x))}$$

Recall that in Theorem 1, we can get the solution $\pi_r(z_i|x) = \frac{\pi_{\mathrm{ref}}(z_i|x) \exp\left(\frac{1}{\beta} r(x, z_i)\right)}{\sum_{j=1}^{N} \pi_{\mathrm{ref}}(z_j|x) \exp\left(\frac{1}{\beta} r(x, z_j)\right)}$, $\pi_{r^*}(z_i|x) = \frac{\pi_{\mathrm{ref}}(z_i|x) \exp\left(\frac{1}{\beta} r^*(x, z_i)\right)}{\sum_{j=1}^{N} \pi_{\mathrm{ref}}(z_j|x) \exp\left(\frac{1}{\beta} r^*(x, z_j)\right)}$. Therefore, the KL divergence between the policy distributions can be written as:

$$\mathbb{D}_{\mathrm{KL}}(\pi_r(z|x)||\pi_{r^*}(z|x))$$

$$= \sum_{i=1}^{N} \pi_r(z_i|x) \log \frac{\pi_r(z_i|x)}{\pi_{r^*}(z_i|x)}$$

$$= \sum_{i=1}^{N} \pi_r(z_i|x) \log \frac{\frac{\pi_{\mathrm{ref}}(z_i|x) \exp\left(\frac{1}{\beta} r(x, z_i)\right)}{\sum_{j=1}^{N} \pi_{\mathrm{ref}}(z_j|x) \exp\left(\frac{1}{\beta} r(x, z_j)\right)}}{\frac{\pi_{\mathrm{ref}}(z_i|x) \exp\left(\frac{1}{\beta} r^*(x, z_i)\right)}{\sum_{j=1}^{N} \pi_{\mathrm{ref}}(z_j|x) \exp\left(\frac{1}{\beta} r^*(x, z_j)\right)}}$$

$$= \sum_{i=1}^{N} \pi_r(z_i|x) \left[ \log \exp\left(\frac{1}{\beta}(r(x, z_i) - r^*(x, z_i))\right) - \log \frac{\sum_{j=1}^{N} \pi_{\mathrm{ref}}(z_j|x) \exp\left(\frac{1}{\beta} r(x, z_j)\right)}{\sum_{j=1}^{N} \pi_{\mathrm{ref}}(z_j|x) \exp\left(\frac{1}{\beta} r^*(x, z_j)\right)} \right]$$

$$= \sum_{i=1}^{N} \pi_r(z_i|x) \left[ \left(\frac{1}{\beta}(r(x, z_i) - r^*(x, z_i))\right) \right.$$

$$\left. - \log \frac{\sum_{j=1}^{N} \pi_{\mathrm{ref}}(z_j|x) \exp\left(\frac{1}{\beta} r^*(x, z_j)\right) \exp\left(\frac{1}{\beta}(r(x, z_j) - r^*(x, z_j))\right)}{\sum_{j=1}^{N} \pi_{\mathrm{ref}}(z_j|x) \exp\left(\frac{1}{\beta} r^*(x, z_j)\right)} \right]$$

$$= \sum_{i=1}^{N} \pi_r(z_i|x) \left[ \left(\frac{1}{\beta}(r(x, z_i) - r^*(x, z_i))\right) \right.$$

$$\left. - \log \sum_{j=1}^{N} \left( \frac{\pi_{\mathrm{ref}}(z_j|x) \exp\left(\frac{1}{\beta} r^*(x, z_j)\right)}{\sum_{j=1}^{N} \pi_{\mathrm{ref}}(z_j|x) \exp\left(\frac{1}{\beta} r^*(x, z_j)\right)} \right) \exp\left(\frac{1}{\beta}(r(x, z_j) - r^*(x, z_j))\right) \right]$$

$$= \sum_{i=1}^{N} \pi_r(z_i|x) \left[ \frac{1}{\beta}(r(x, z_i) - r^*(x, z_i)) - \log \sum_{j=1}^{N} \pi_{r^*}(z_j|x) \exp\left(\frac{1}{\beta}(r(x, z_j) - r^*(x, z_j))\right) \right]$$

Using Jensen's inequality, we have:

$$- \log \sum_{j=1}^{N} \pi_{r^*}(z_j|x) \exp\left(\frac{1}{\beta}(r(x, z_j) - r^*(x, z_j))\right)$$

$$\leq - \sum_{j=1}^{N} \pi_{r^*}(z_j|x) \log \exp\left(\frac{1}{\beta}(r(x, z_j) - r^*(x, z_j))\right) = - \sum_{j=1}^{N} \pi_{r^*}(z_j|x)\left(\frac{1}{\beta}(r(x, z_j) - r^*(x, z_j))\right)$$

Therefore, we have:

$$\mathbb{D}_{\text{KL}}\left(\pi_r(z|x) || \pi_{r^*}(z|x)\right)$$

$$\leq \sum_{i=1}^{N} \pi_r(z_i|x)\left[\frac{1}{\beta}(r(x, z_i) - r^*(x, z_i)) - \sum_{j=1}^{N} \pi_{r^*}(z_j|x)\left(\frac{1}{\beta}(r(x, z_j) - r^*(x, z_j))\right)\right]$$

$$\leq \sum_{i=1}^{N} \pi_r(z_i|x)\left[\frac{1}{\beta}|r(x, z_i) - r^*(x, z_i)| + \sum_{j=1}^{N} \pi_{r^*}(z_j|x)\left(\frac{1}{\beta}|r(x, z_j) - r^*(x, z_j)|\right)\right]$$

$$\leq \sum_{i=1}^{N} \pi_r(z_i|x)\left[\frac{\epsilon}{\beta} + \frac{\epsilon}{\beta}\sum_{j=1}^{N} \pi_{r^*}(z_j|x)\right]$$

In Theorem 1, we have the constraint that $\sum_{j=1}^{N} \pi_r(z_j|x) = 1$, and $\sum_{j=1}^{N} \pi_{r^*}(z_j|x) = 1$. Therefore, the KL divergence between the policy distributions can be written as:

$$\mathbb{D}_{\text{KL}}\left(\pi_r(z|x) || \pi_{r^*}(z|x)\right) \leq \sum_{i=1}^{N} \pi_r(z_i|x)\left[\frac{\epsilon}{\beta} + \frac{\epsilon}{\beta}\sum_{j=1}^{N} \pi_{r^*}(z_j|x)\right]$$

$$= \sum_{i=1}^{N} \pi_r(z_i|x)\left[\frac{\epsilon}{\beta} + \frac{\epsilon}{\beta}\right] = 1 \cdot \frac{2\epsilon}{\beta} = \frac{2\epsilon}{\beta}$$

Putting all the results together, we get:

$$|\mathbb{E}_{z \sim \pi_r(z|x)} r^*(x, z) - \mathbb{E}_{z \sim \pi_{r^*}(z|x)} r^*(x, z)| \leq \sqrt{\frac{4\epsilon}{\beta}}$$

This theorem establishes a bound on the expected correctness rate of trajectories $z$ generated using the trained LRM in comparison to the perfect reward model. As the performance of the classifier improves, the error $\epsilon$ will drop, leading to a tighter bound and higher expected correctness rate. Notably, even if the latent policy of the base model is not explicitly optimized, a more accurate LRM with a smaller $\epsilon$ enables LTO to more accurately select only the correct latent thinking trajectories, thereby improving the expected correctness rate. Empirically, as shown in Section 3.3, the classifier achieves a very high AUC-ROC, implying that $\epsilon$ is small in practice. From a theoretical perspective, standard generalization bounds for binary classifiers guarantee that the reward error $\epsilon$ is controlled by the classification error on the training set plus a complexity term of order $O(\sqrt{1/S})$ with $S$ being the number of training samples (Bartlett & Mendelson, 2002; Bartlett et al., 2017). Consequently, with a well-trained classifier as the reward model, this bound guarantees that the expected correctness rate under the trained reward model closely matches that of the perfect reward model.

## G EXPERIMENTAL DETAILS

### G.1 DATASET DETAILS

We select five datasets from three domains for a comprehensive evaluation. The details of datasets are described as follows:

• *Math Problems*

  • **GSM8K** (Cobbe et al., 2021) is a collection of grade-school math word problems written by human annotators. The dataset is designed to evaluate arithmetic and reasoning skills at the

grade-school level and serves as a benchmark for testing the multi-step reasoning capability of LLMs. It is divided into 7,473 training problems and 1,318 test problems, and each problem is paired with a detailed step-by-step solution based on basic arithmetic operations.

- **GSM-Symbolic** (Mirzadeh et al., 2025) is a more challenging extension of GSM8K that generates diverse math problem variants using symbolic templates. It includes 5,000 test problems but does not provide a training split.

- **SVAMP** (Patel et al., 2021) is also a collection of grade-school math word problems. It is constructed by applying systematic variations to seed examples from the ASDiv dataset (Miao et al., 2020) to discourage shortcut reasoning patterns. The dataset is split into 35,381 training problems and 1,000 test problems.

- *Commonsense Reasoning*

  - **CommonsenseQA** (Talmor et al., 2019) is a multiple-choice question answering benchmark dataset designed to evaluate the capability of LLMs to perform commonsense reasoning. It consists of 9,741 training problems and 1,221 test problems.

- *Code Generation*

  - **MBPP** (Austin et al., 2021) is a benchmark dataset for evaluating the capability of LLMs to generate programming codes. It consists of Python programming problems covering basic algorithmic and data-processing tasks. Each problem is paired with a natural language description, a reference implementation, and multiple test cases. The generated code is considered correct only if it successful passes all the test cases. The dataset is divided into 374 training problems and 483 test problems.

## G.2 IMPLEMENTATION DETAILS

To train the latent classifier as the LRM, we generate multiple latent thinking trajectory-answer pairs for each dataset. For GSM8K, SVAMP, and CommonsenseQA, we sample 5 different latent thinking trajectories and answers per problem from the training split. For MBPP, which contains only 373 training problems, we sample 50 latent thinking trajectories and answers per problem to ensure sufficient training data. The latent classifier is trained to predict the correctness of the answer from the latent thoughts on each dataset. For GSM-Symbolic, which does not include a training split, we use the classifier trained on GSM8K. We evaluate the performance of baselines and our approach on the test split of each dataset. For LTO and the baselines, we allocate a sampling budget of $N{=}20$ per problem. Each method selects a single final solution from these candidates (i.e., the number of required samples $M{=}1$). For baselines, the solution with the highest evaluation score (e.g., verbal evaluation score, confidence score or CoE score) will be selected. We adopt the default setting with sampling budget $N{=}20$ and latent thinking steps $T{=}32$, and the performances with different sampling budget, different $\beta$s and different number of thinking steps are studied in Section H.2, Section H.3 and Section H.4, respectively.

For LRMs on general LLMs, we follow the same training configuration as in Appendix E. For each LLM, we configure the corresponding LRM with the same hidden dimensionality, number of attention heads, and MLP hidden size as that LLM, following the training setup in Appendix E. For example, if an LLM has hidden dimensionality 4096, number of attention heads 32, and MLP hidden size 14336, then the LRM also has the same hidden dimensionality 4096, attention heads 32 and MLP hidden size 14336. The hidden states from all layers are stacked together to form a sequence of latent thoughts. For example, the hidden representation from layer 1 is treated as latent thought step 1, the representation from layer 2 as step 2, and so on. This stacked sequence of latent thoughts from general LLMs serves as the input to the LRM, and it has exactly the same format as the sequences of latent thoughts from Huginn-3.5B. To ensure that general LLMs will generate different latent representations for multiple samples of latent representations, we randomly sample one example (problem-answer pair) from the training split of each dataset, and append this example as an in-context demonstration to the input question. For GSM-Symbolic, which lacks a training set, we instead draw examples from the training set of GSM8K. Because a new example is drawn at each iteration, the input tokens and consequently the latent representations will be different across multiple samples.

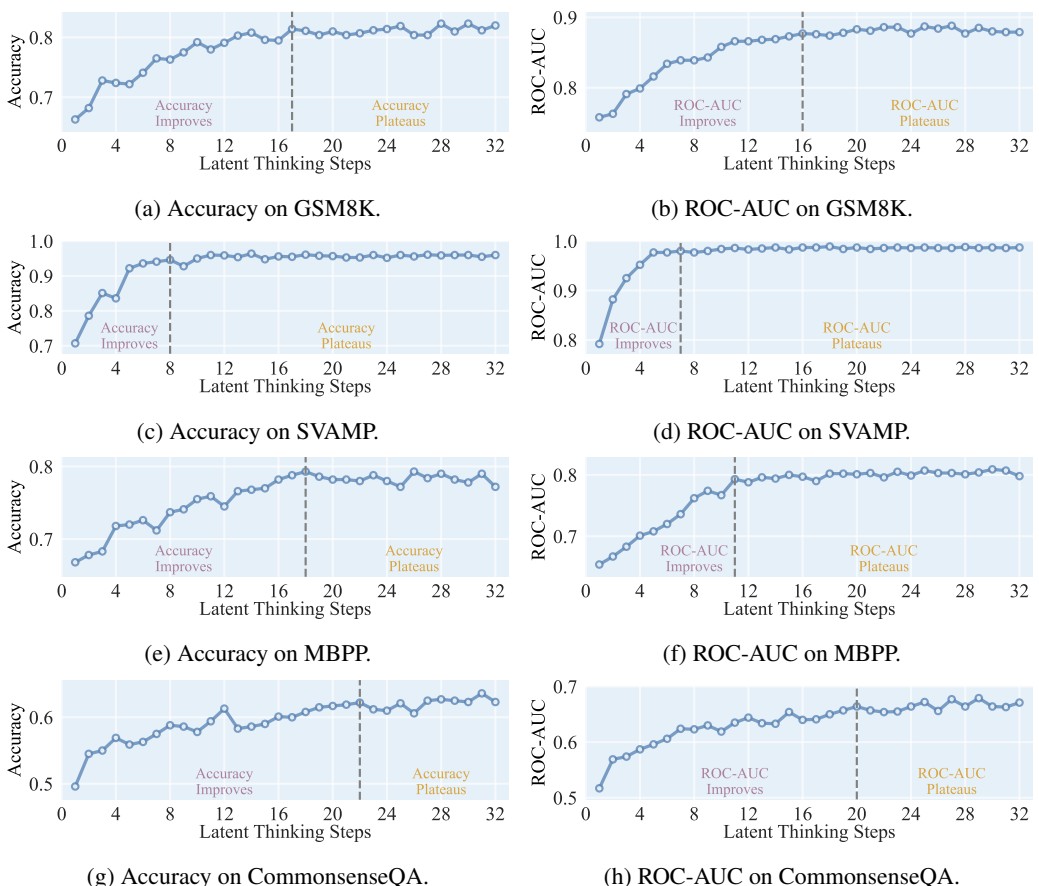

Figure A3: Test-set performance of the latent classifier (measured by Accuracy and ROC-AUC) on the test set trained with varying numbers of latent thinking steps on the SVAMP and MBPP datasets.

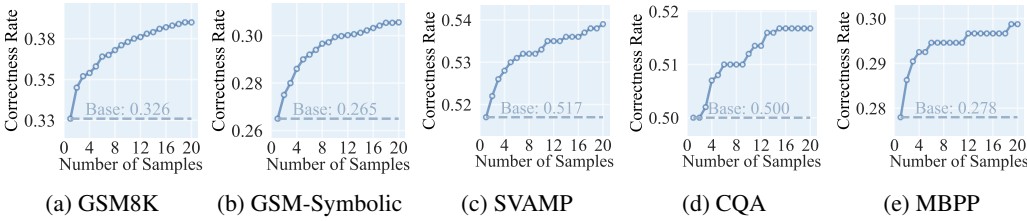

Figure A4: Performance of LTO with different numbers of samples. "CQA" refers to the CommonsenseQA dataset. "Base" refers to the performance of the base model.

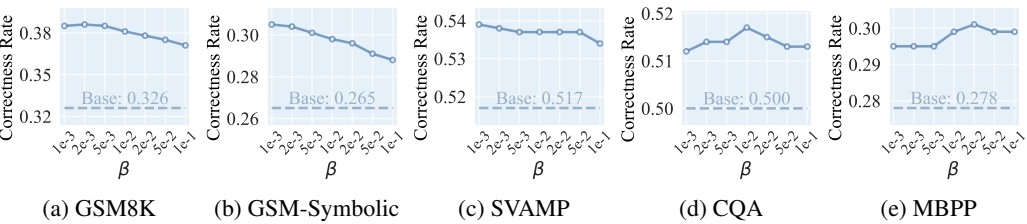

Figure A5: Performance of LTO with different betas. "CQA" refers to the CommonsenseQA dataset. "Base" refers to the performance of the base model.

Table A2: Performance of the latent classifier on the test set for general LLMs on different datasets.

| Model | Metric | GSM8K | SVAMP | CommonsenseQA | MBPP |
|---|---|---|---|---|---|
| OLMo-7B | Accuracy | 0.896 | 0.854 | 0.652 | 0.858 |
| | ROC-AUC | 0.851 | 0.899 | 0.708 | 0.882 |
| Llama-2-7B | Accuracy | 0.836 | 0.919 | 0.681 | 0.834 |
| | ROC-AUC | 0.858 | 0.970 | 0.738 | 0.822 |
| Llama-2-13B | Accuracy | 0.805 | 0.925 | 0.729 | 0.805 |
| | ROC-AUC | 0.868 | 0.974 | 0.773 | 0.839 |
| Mistral-7B | Accuracy | 0.793 | 0.968 | 0.736 | 0.741 |
| | ROC-AUC | 0.868 | 0.992 | 0.765 | 0.794 |

## H ADDITIONAL EXPERIMENTAL RESULTS

### H.1 ADDITIONAL RESULTS ON THE PERFORMANCE OF THE LATENT CLASSIFIER

Additional experimental results on the performance of latent classifier for Huginn-3.5B using different thinking steps on different datasets are shown in Figure A3.

Additional experimental results on the performance of latent classifier for general LLMs on different datasets are shown in Table A2. In this setting, each LRM is trained with the latent representations from all the layers of each general LLM.

We can see that the latent classifier achieves strong performance on the test set for Huginn-3.5B and general LLMs across diverse datasets. These results demonstrate that latent thoughts encode appropriate reward signals that can indicate whether they will lead to the correct answer.

### H.2 PERFORMANCE WITH DIFFERENT SAMPLING BUDGET

To investigate the performance of LTO with different sampling budget $N$, we vary $N$ from 1 to 20 and report the performance of LTO in Figure A4. Performance steadily improves as $N$ increases, as a larger $N$ enhances the diversity of sampled latent thoughts and increases the likelihood that at least one sampled latent thinking trajectory is correct. Moreover, even with a very small budget (e.g., $N = 2$), LTO can still achieve substantial performance improvement compared with the base model, demonstrating that LTO is sample-efficient without the need for a large sampling budget.

### H.3 PERFORMANCE WITH DIFFERENT BETAS

To investigate the performance of LTO with different $\beta$, we vary $\beta$ from $1e-3$ to $1e-1$ and report the performance of LTO in Figure A5. Across different values of $\beta$, LTO consistently outperforms the base model, demonstrating that it can reliably improve the latent thinking processes with different choices of the hyperparameter.

### H.4 PERFORMANCE WITH DIFFERENT NUMBERS OF THINKING STEPS

While most of our evaluation uses a fixed number of latent thinking steps, we also investigate the adaptability of LTO to latent thinking trajectories of varying thinking steps. Specifically, for each dataset, we train the LRM with the sampled latent thinking trajectories with varying number of thinking steps. We then test the performance of LTO using this LRM trained with varying number of thinking steps.

From the experimental results in Table A3, we can see that LTO achieves a consistent improvement over the base model in all the cases, indicating that LTO can be flexibly applied to latent thinking trajectories of varying numbers of thinking steps. Interestingly, performance slightly declines as the number of steps increases. This is attributed to the reduced diversity in the sampled latent thoughts and answers when longer thinking steps are used. For example, on SVAMP, when using 16 thinking steps, 427 problems have sampled answers that are all incorrect, 437 problems have sampled answers that are all correct, and 136 problems have both correct and incorrect answers.

Table A3: Performance of LTO with different numbers of thinking steps. For each thinking step, the best-performing method is highlighted in **bold**. ∗ indicates the improvement over the best runner-up is statistically significant with $p < 0.05$.

| Thinking Steps | Method | GSM8K | GSM-Symbolic | SVAMP | CommonsenseQA | MBPP |
|---|---|---|---|---|---|---|
| 16 Steps | Base Model | 0.333 | 0.269 | 0.503 | 0.498 | 0.276 |
| | Majority Voting | 0.345 | 0.279 | 0.501 | 0.498 | 0.274 |
| | Latent Thinking Optimization | **0.434**∗ | **0.335**∗ | **0.560**∗ | **0.523**∗ | **0.295**∗ |
| 24 Steps | Base Model | 0.326 | 0.265 | 0.515 | 0.507 | 0.282 |
| | Majority Voting | 0.334 | 0.274 | 0.513 | 0.509 | **0.293** |
| | Latent Thinking Optimization | **0.398**∗ | **0.312**∗ | **0.549**∗ | **0.523**∗ | 0.293 |
| 32 Steps | Base Model | 0.326 | 0.265 | 0.517 | 0.500 | 0.278 |
| | Majority Voting | 0.333 | 0.269 | 0.511 | 0.504 | 0.288 |
| | Latent Thinking Optimization | **0.378**∗ | **0.303**∗ | **0.539**∗ | **0.520**∗ | **0.295**∗ |

Table A4: Performance comparison of the Llama model family. The best-performing method for each model is in **bold**. ∗ indicates the improvement over the best runner-up is statistically significant with $p < 0.05$.

| Model | Method | GSM8K | GSM-Symbolic | CommonsenseQA | MBPP |
|---|---|---|---|---|---|
| Llama-2-7B | Base Model | 0.223 | 0.204 | 0.399 | 0.189 |
| | Majority Voting | 0.275 | 0.302 | 0.493 | 0.193 |
| | Latent Thinking Optimization | **0.389**∗ | **0.316**∗ | **0.606**∗ | **0.237**∗ |
| Llama-2-13B | Base Model | 0.306 | 0.273 | 0.398 | 0.247 |
| | Majority Voting | 0.417 | 0.379 | 0.501 | 0.263 |
| | Latent Thinking Optimization | **0.534**∗ | **0.442**∗ | **0.650**∗ | **0.322**∗ |
| Llama-3-8B | Base Model | 0.784 | 0.736 | 0.742 | 0.560 |
| | Majority Voting | 0.801 | 0.796 | 0.786 | 0.570 |
| | Latent Thinking Optimization | **0.859**∗ | **0.821**∗ | **0.790**∗ | **0.600**∗ |

Therefore, the performance upper bound is $(437 + 136)/1000 = 0.573$. By comparison, when using 24 thinking steps, the split becomes $446/468/86$ with the performance upper bound calculated as $(468 + 85)/1000 = 0.554$; when using 32 thinking steps, the split becomes $454/486/60$ with the performance upper bound calculated as $(486 + 60)/1000 = 0.546$.

While increasing the number of thinking steps slightly improves the expected correctness rate of the base model, it substantially reduces the diversity of sampled latent thoughts and answers, probably due to overthinking (Sui et al., 2025). As a result, fewer problems contain both correct and incorrect answers (i.e., diverse sets), leaving less room for improvement with LTO. It is possible that there exists an optimal number of thinking steps that balances the expected correctness rate of the base model with the diversity of the latent thoughts and answers, and future work may design adaptive mechanisms to identify such optimal thinking steps and further improve the performance of LTO.

## H.5 PERFORMANCE COMPARISON ON LLM MODEL FAMILY

To further validate the effectiveness of LTO on general LLMs, we conduct an additional experiment evaluating LTO on the widely-used Llama model family (Touvron et al., 2023b; Llama Team, 2023). The experimental results in Table A4 show that LTO consistently enhances the reasoning performance across all models, demonstrating its effectiveness in improving the latent thinking processes of the LLM model family.

## H.6 PERFORMANCE ON MORE CHALLENGING BENCHMARKS

To further validate the effectiveness of LTO on general LLMs, we provide an additional analysis using two more recent LLMs (Llama-3-8B (Llama Team, 2023) and Qwen-3-4B (Yang et al., 2025)) on two broader, frontier benchmarks (MATH (Hendrycks et al., 2021) and GPQA (Rein et al., 2024)),

Table A5: Performance of LTO on more challenging benchmarks. The best-performing method for each model is in **bold**. $*$ indicates statistically significant improvement with $p < 0.05$.

| Model | Method | MATH | GPQA |
|---|---|---|---|
| | Base Model | 0.267 | 0.268 |
| Llama-3-8B | Majority Voting | 0.335 | 0.276 |
| | Latent Thinking Optimization | **0.375**$^*$ | **0.310**$^*$ |
| | Base Model | 0.552 | 0.347 |
| Qwen-3-4B | Majority Voting | 0.555 | 0.368 |
| | Latent Thinking Optimization | **0.619**$^*$ | **0.490**$^*$ |

Table A6: Comparison of the total training time and GPU memory usage of LRM across different datasets and settings. "General" denotes the general reward model from Section 6.3.

| | GSM8K | SVAMP | CommonsenseQA | MBPP | General |
|---|---|---|---|---|---|
| Total Training Time (h) | 0.85 | 4.19 | 1.05 | 0.92 | 6.12 |
| GPU Memory Usage (GB) | 10.39 | 10.40 | 10.39 | 10.40 | 10.39 |

which are known to be relatively noisy and pose more challenging reasoning conditions for LLMs. The results in Table A5 show that LTO consistently enhances the reasoning performance across all models and datasets, demonstrating its effectiveness and robustness in improving the latent thinking processes of the general LLMs on broader benchmarks under noisy and challenging conditions.

## H.7   EFFICIENCY ANALYSIS

We evaluate the efficiency of our framework from two perspectives: the training efficiency of LRM and the sampling efficiency of LTO. Our results demonstrate that LRM requires only modest resources to train, and sampling answers with LTO brings negligible additional cost during inference.

**Training Efficiency of LRM**   We analyze the training efficiency of the LRM on Huginn-3.5B by measuring the total training time and GPU memory usage of LRM across different datasets and settings. All the experiments are conducted on a single A100 GPU using the default 32 thinking steps. From the experimental results in Table A6, we can see that the training of LRM can be completed within reasonable time and modest memory budgets in all the settings. Such resource cost is significantly lower than that of language-based reward models (Wang et al., 2024; Lu et al., 2024). These results demonstrate that reward modeling in the latent space offers a more efficient alternative to reward modeling in the natural language space.

**Sampling Efficiency of LTO**   Compared to standard inference procedure, which directly samples latent thoughts and responses from the base model, LTO introduces an additional step for latent reward computation. To evaluate the efficiency of this step on Huginn-3.5B, we compare the average computation time of the base model inference and the latent reward computation per sample across five datasets. All the experiments are conducted on a single A100 GPU using the default 32 thinking steps. From the experimental results in Table A7, we can see that the computation time of LRM is orders of magnitude lower than the inference time of the base model, indicating that LRM is highly efficient and incurs little computation cost. Moreover, since there are not sequential dependencies between the sampled latent thinking trajectories, the sampling process in LTO can be fully parallelized. Therefore, LTO incurs only negligible additional inference cost, and its total inference time can be almost the same with direct sampling from the base model when parallel sampling is introduced.

**Efficiency Analysis on General LLMs**   LRMs are highly efficient both for general LLMs and Huginn-3.5B due to its lightweight architecture with only 2 layers of Transformer encoder. Its training time is small because it only requires a small amount of training data and its inference time is orders of magnitude lower than the inference time of the base LLM. For example, as shown in

Table A7: Comparison of the average computation time (seconds) of the base model inference and the latent reward computation per sample across five datasets.

|  | GSM8K | GSM-Symbolic | SVAMP | CommonsenseQA | MBPP |
|---|---|---|---|---|---|
| Based Model Inference | 39.5 | 43.0 | 6.0 | 7.3 | 20.4 |
| Latent Reward Computation | 7.6e-02 | 7.6e-02 | 7.5e-02 | 7.5e-02 | 7.6e-02 |

Table A8: Comparison of the total training time and GPU memory usage of LRM across different datasets for Llama-3-8B.

|  | GSM8K | CommonsenseQA | MBPP | MATH | GPQA |
|---|---|---|---|---|---|
| Total Training Time (h) | 1.21 | 1.53 | 0.66 | 1.69 | 0.42 |
| GPU Memory Usage (GB) | 6.39 | 6.40 | 6.39 | 6.39 | 6.39 |

Table A6 and Table A9, for Llama-3-8B, on a single A100 GPU, LRM only requires about 1.5 hrs to train and 6.4GB of memory usage, and its reward computation can be completed within about 0.02s which is negligible compared with the inference time of the base LLM.

## I  LIMITATION STATEMENT

**Direct Optimization over the Latent Thinking Processes**  Although LTO is formulated as an optimization problem, it achieves the optimization objective by selectively sampling correct latent thinking trajectories that follow the optimized distribution, rather than directly modifying or updating the latent thinking policy of the base model. As with the common limitation of test-time scaling methods (Gandhi et al., 2025; Setlur et al., 2025), when the latent thinking policy of the base model diverges substantially from the optimized distribution (e.g., when the model lacks the problem-solving ability and generates latent thoughts and answers that are all incorrect), then LTO cannot improve the latent thinking processes, since every generated latent thinking trajectory remains incorrect. To address this limitation, future works may integrate the reward signals from LRM into a reinforcement learning-based preference optimization framework (Rafailov et al., 2023), enabling direct optimization and refinement of the latent thinking processes.

**Optimization with Multiple Reward Signals**  The reward signals derived from LTO are binary and can only indicate if the latent thinking processes will lead to the correct answer. This restricts reward modeling to the correctness of the answer but may not capture other important dimensions, such as safety or helpfulness. An interesting direction for future work is to investigate whether latent thoughts are separable along these additional dimensions, and to extend the latent classifier to incorporate these criteria for latent reward modeling. Another interesting direction is to extend LTO into a multi-objective optimization framework (Wang et al., 2025a). This will enable simultaneous optimization of latent thinking processes across multiple reward dimensions and broaden its applicability to more general settings for reward optimization and preference alignment.

## J  IMPACT STATEMENT

Existing methods for reward modeling and thinking optimization in LLMs are primarily performed in the natural language space (Wang et al., 2024; Lu et al., 2024), but may be costly and prone to the overthinking issue (Sui et al., 2025). Our research demonstrates that the latent representations of both Huginn-3.5B and general LLMs encode appropriate reward signals that can be directly leveraged to optimize the latent thinking processes. Furthermore, we show that reward modeling in the latent space can generalize across domains and shows strong potential for building a generalist reward model in the latent space. Our results demonstrate that reward modeling and scaling test-time thinking with supervision (Muennighoff et al., 2025; Guo et al., 2025; Setlur et al., 2025) can be performed directly in the latent space, highlighting its potential as a general, efficient, and domain-agnostic approach to improving the thinking processes of LLMs.

Table A9: Comparison of the average computation time (seconds) of the base model inference and the latent reward computation per sample across different datasets for Llama-3-8B.

|  | GSM8K | CommonsenseQA | MBPP | MATH | GPQA |
|---|---|---|---|---|---|
| Based Model Inference | 8.5 | 8.2 | 3.0 | 12.5 | 21.8 |
| Latent Reward Computation | 1.3e-2 | 1.6e-2 | 5.0e-3 | 5.0e-3 | 5.0e-3 |

We do not aim to claim that latent reward modeling and latent thinking optimization are better than natural language-based reward modeling and verbal thinking optimization. Instead, we show that they offer efficient and effective alternatives in specific settings and open up promising directions for future works. For example, in resource-constrained settings where training computation is limited and inference efficiency is imperative, LTO can effectively optimize the thinking processes of LLMs with low computation cost. We hope that our research motivates further exploration of reward modeling and thinking optimization in the latent space—a largely underexplored but highly promising direction for advancing scalable, efficient, and generalist LLM thinking and reasoning.

## K    ETHICS STATEMENT

All the datasets used in this research are from public open-access benchmark datasets, which are fully anonymized and do not contain sensitive or private information.

## L    REPRODUCIBILITY STATEMENT

Calculation methods for the representation quality methods are provided in Appendix C. A complete proof of the theorems is provided in Appendix F. The implementation details and the computational cost of the LRMs are provided in Appendix E and Appendix H.7, respectively. The implementation details of the baseline methods are provided in Appendix G.2. Our code and datasets are available at this link.

## M    USAGE OF LARGE LANGUAGE MODEL

We do not use large language models as contributors to generate any part of the content or write the paper. Large language models are only used as the investigation object of our study (e.g., observe how large language models think in the latent space).

