# OpenReview forum: "Latent Thinking Optimization: Your Latent Reasoning Language Model Secretly Encodes Reward Signals in Its Latent Thoughts"
_ICLR.cc/2026/Conference — ICLR 2026 Poster_

### Official Review · Reviewer_qGuA · 2025-10-31

**Soundness:** 3
**Presentation:** 3
**Contribution:** 3
**Rating:** 6
**Confidence:** 4

**Summary:**

The paper studies latent thinking in the latent-reasoning LM Huggin-3.5B and claims that correctness signals are encoded in latent trajectories. The authors (i) empirically show separable patterns between correct vs. incorrect latent thoughts via visualization and four representation metrics (entropy, effective rank, anisotropy, intrinsic dimension); (ii) train a latent classifier that predicts answer correctness from partial latent trajectories with high ROC-AUC; and (iii) propose Latent Thinking Optimization (LTO): use the classifier as a Latent Reward Model (LRM) and perform KL-regularized probabilistic selection of latent trajectories via an acceptance–rejection sampler with proofs (Thm. 1/2). LTO improves correctness over voting/self-correction and latent heuristics and is further applied to general LLMs by treating intermediate layer states as “latent thoughts,” showing sizable gains and cross-domain transfer of the LRM.

**Strengths:**

1. neat empirical evidence that correct trajectories differ from incorrect ones, with stepwise analyses and a probing classifier that improves with longer prefixes.

2. LTO formulates a KL-regularized objective and provides a closed-form reweighting plus an accept-reject sampler with correctness guarantees (Thm. 1/2).

3. Broad task coverage & signal utility: improvements across math/commonsense/code; LRM used as weights already helps; LTO helps more.

4. Generality claims with initial evidence: application to general LLMs (OLMo/Llama/Mistral) and cross-dataset LRM transfer; training-data footprint noted as modest.

**Weaknesses:**

1. Entropy/effective-rank/anisotropy/intrinsic-dimension are scale- and whitening-sensitive; without controlling for per-step activation scaling or layer-norm statistics, a change in metric may reflect variance rescaling rather than “better thinking.” Please specify invariances and show robustness across metric definitions and layers.

2. The LRM is trained from the same model’s latent trajectories using final-answer correctness as labels; if the generator exhibits systematic artifacts (e.g., decoding shortcuts at later steps), the classifier may learn spurious correlates. The strong AUC near the end steps could partially capture answer-formation traces rather than “process quality.” A control where latents are early-stopped and the answer is masked/not decoded would help.

3.  The reward signa is derived from LTO are binary and can only indicate if the latent thinking processes will lead to the correct answer, as you mentioned in appendix

**Questions:**

1. As mentioned in Author's guidence, you should disclose the use of Large Language Models. I don't find it in your paper. Could you disclose it here? or you may violate the guidence and lead to desk rejection.

2. Please see weaknesses and try to answer 1-2. For 3, I am curious about methods you have tried.

---

> ### Author Response · Authors · 2025-11-19
>
> ## W1. Rescaling of latent representation metrics
>
> Thanks for the comment! In the Huginn-3.5B architecture, the same RMSNorm module with the same rescaling weight w is applied to each step of latent thought. Therefore, all the latent thoughts from different steps are already normalized to the same scale before we calculate the representation quality metrics, making these latent representations scale-invariant. We will clarify this point in the revision.
>
> ## W2. A control where latents are early-stopped and the answer is masked/not decoded would help.
>
> Thanks for the comment! In Table A1 on page 22, we have an ablation study where we train the LRM with only the latent thoughts corresponding to the first 10 decoded tokens (these tokens almost correspond to early reasoning steps and occur before any answer tokens or final-stage decoding patterns appear) and the last 10 decoded tokens (these tokens almost correspond to the decoded answer tokens). Even under this strict early-stopping condition with only first 10 tokens, the LRM still achieves strong classification accuracy comparable to default setting with all the tokens. This demonstrates that the LRM is not merely exploiting superficial answer-formation traces and is learning meaningful patterns about the quality of the latent thinking processes. While the last-10-tokens variant does show slightly higher AUC as expected, since it contains almost all the answer information, the first-10-tokens results provide clear evidence that the LRM captures process-level correctness signals and does not rely solely on late-stage artifacts.
>
> ## W3. The reward signal is binary.
>
> Thanks for the comment! The design of the reward signal follows the widely adopted Reinforcement Learning with Verifiable Reward (RLVR) paradigm [1], where a binary reward indicates whether the latent thinking process leads to the correct answer, which is standard practice in most LLM RL papers. As we note in Appendix H, we acknowledge that real-world applications may require complex reward signals beyond answer correctness. In our experiments, we observe that latent thoughts can also be separated along other dimensions, such as whether LLMs are generating hallucinated content, or what types of thinking (e.g., reflection or direct thinking) LLMs are engaging in. Exploring these directions can open the door for future research on reward modeling in the latent space, where latent thoughts can be optimized with more complex, possibly multi-objective reward functions measuring different aspects of the latent thinking processes, as discussed in Appendix H.
>
> Reference
>
> [1] Guo, Daya, et al. "Deepseek-r1 incentivizes reasoning in llms through reinforcement learning." Nature 645.8081 (2025): 633-638.
>
> ## Q1. Use of large language models
>
> Thanks for the comment! We do not use large language models as contributors to generate any part of the content or write the paper. Large language models are only used as the investigation object of our study (e.g., observe how large language models think in the latent space). We will clarify this point in the revision.

---

### Official Review · Reviewer_jnr6 · 2025-11-01

**Soundness:** 3
**Presentation:** 4
**Contribution:** 2
**Rating:** 6
**Confidence:** 3

**Summary:**

This paper studies how reasoning language models internally represent their “thoughts” in latent space. Using the Huggin-3.5B model, the authors show that hidden-state trajectories corresponding to correct and incorrect answers form clearly separable patterns. They train a small latent classifier to detect these patterns and use it as a Latent Reward Model (LRM) in a new method called Latent Thinking Optimization (LTO)—a test-time sampling procedure that reweights latent trajectories toward those predicted to be correct. Without retraining the base model, LTO consistently improves reasoning accuracy across five benchmarks and generalizes to other LLMs (e.g., Llama-2, Mistral), demonstrating that latent representations implicitly encode reward-like signals useful for optimizing reasoning.

**Strengths:**

* The author conduct in-depth analysis of the latent reasoning trajectories, both qualitatively and quantitatively through metrics including entropy, effective rank, anisotropy, and intrinsic dimension.
* The proposed method is simple that only involves computing evaluation metrics as input features to a small LRM, which is later used to guide the selection of answers through LTO.
* The paper is well organized, balancing theoretical exposition with extensive experiments and visualizations.

**Weaknesses:**

* Optimization vs. resampling: LTO is effectively a weighted rejection sampling procedure, not true optimization of the model’s latent policy; the optimization problem formulation is confusing. Also, it not clear if the KL divergence constraint is actually needed.
* Limited interpretability: Despite excellent quantitative separation, there is little qualitative analysis linking specific latent dimensions or manifolds to interpretable reasoning concepts.
* Minor overclaiming: Phrases like “generalist reward model” or “domain-agnostic optimization” are ambitious relative to the experimental evidence.

**Questions:**

Q1: can author explain the KL divergence term. This is is common in LLM post-training, but LTO is essentially not a optimization method, the distribution won't deviate from base policy a lot.
Q2: can author provide more insight about why correct and incorrect latent trajectories lead to different patterns? For example, why correct latent trajectories tend to be more compact dispersed?

---

> ### Author Response · Authors · 2025-11-19
>
> We are grateful that reviewer jnr6 thinks that our research has in-depth analysis, the proposed method is simple and effective with theoretical guarantees and empirical validations. We sincerely appreciate the questions and would like to offer further clarifications:
>
> ## W1-1. Formulation of the optimization problem
>
> Thanks for the comment. We are happy to clarify that the formulation of the optimization problem is theoretically grounded: the goal is to find a good latent policy such that it maximizes the expectation of generating a correct latent thinking trajectory under a KL-regularized constraint over latent trajectory distributions. As proved in theorem 1, the solution to this optimization problem is Equation 2, and the weighted sampling algorithm is an implementation of the solution to this discrete distribution optimization problem. While it does rely on sampling instead of parameter update, we believe that this does not diminish its nature as an optimization problem over the latent policy. This formulation is also consistent with other test-time scaling methods such as Textgrad [1], which formulate optimization objectives over LLM outputs even though no LLM parameters are updated. We agree that this formulation might cause some confusion given there is no parameter actually updated, and we will better clarify this point in the revision.
>
> Reference:
> [1] Yuksekgonul, Mert, et al. "Optimizing generative AI by backpropagating language model feedback." Nature 639.8055 (2025): 609-616.
>
>
> ## W1-2 & Q1. The KL divergence weight
>
> Thanks for the comment! We agree that the necessity of the KL regularization term beta might not seem obvious, but we believe that our theoretical and empirical results support that beta is necessary.
>
> The inclusion of beta is well motivated from the KL-regularized policy optimization theory. Without KL regularization, the optimized policy might collapse into a degenerated policy that significantly deviates from the base policy. Moreover, if we remove beta (set beta→0), LTO will reduce to best-of-n sampling in latent space, i.e., we sample N latent trajectories, score them with the LRM, and pick the highest-scoring one. This strategy only works well if the LRM is nearly perfect with almost 1.0 classification accuracy for correct and incorrect latent trajectories. In practice, the LRM is learned and not perfect. Without any regularization, LTO may exploit the errors of the LRM and select suboptimal latent thoughts. By including a KL penalty, we constrain the optimized policy so that it does not drift too far from the base policy, thereby preserving the diversity of sampled latent thoughts and mitigating overfitting to reward model noise.
>
> In our experiments, we also observe that removing beta shows worse performance than using a moderate beta value. For example, in the following experiments on Mistral-7B, when we push beta→0, performance consistently drops across different tasks. In contrast, tuning beta to a modest nonzero value yields better results. Even though our setting is test-time scaling where the policy deviation may not be as large as in full RL fine-tuning, we still see worse performance when beta→0 is zero. This confirms that KL regularization is required to achieve optimal results with LTO.
>
> | beta  | GSM8K    | GSM-Symbolic | SVAMP|CommonsenseQA       | MBPP       |
> |---------------|-----------------------------|------------|-----------|------------|-----------|
> | beta→0     | 0.530 | 0.450  | 0.760     | 0.690     | 0.376
> |    optimal beta         | 0.565 (beta = 0.005) | 0.461 (beta = 0.01)  | 0.771 (beta=0.02)     | 0.708 (beta = 0.07) |0.388 (beta = 0.002)

---

> ### Author Response · Authors · 2025-11-19
>
> ## W2. Interpretability of latent thoughts
>
> Thanks for the comment! Here we provide an interpretability analysis by decoding the latent thoughts at different reasoning steps and examining how they evolve toward (or away from) the correct answer. To make this analysis clear, we use a one-digit arithmetic dataset, where the model must output a single digit answer to an arithmetic question. This setting is ideal for interpretability because the operations are simple and easy to verify, and we can directly inspect how the decoded latent thoughts evolve at the specific token position corresponding to the answer digit. Using the coda module (decoder) from the latent reasoning language model, we decode the latent vector at each thinking step into its top-5 most probable tokens and analyze the progression of latent thoughts. Below we show representative examples with correct and incorrect thinking patterns.
>
> Examples with correct thinking patterns:
>
> Example 1 (Answer: 7):
> ```
> Question: What is (1 * 1) + 6?
> Answer:
> ```
> Top-5 ranked tokens decoded from latent thoughts at different thinking steps:
> ```
> Step 16: ['-', '1', '3', '2', '4']
> Step 32: ['1', '7', '6', '2', '8']
> Step 48: ['7', '6', '1', '8', '2']
> Step 64: ['7', '6', '1', '8', '2']
> ```
>
> Example 2 (Answer: 5):
> ```
> Question: What is (7 + 2) - 4?
> Answer:
> ```
>
> Top-5 ranked token decoded from latent thoughts at different thinking steps:
> ```
> Step 16: ['-', '1', '3', '2', '4']
> Step 32: ['1', '5', '9', '2', '6']
> Step 48: ['5', '1', '-', '7', '4']
> Step 64: ['5', '7', '9', '1', '-']
> ```
> Example 3 (Answer: 7):
> ```
> Question: What is (3 * 2) + 1?
> Answer:
> ```
> Top-5 ranked token decoded from latent thoughts at different thinking steps:
> ```
> Step 16: ['-', '1', '3', '2', '4']
> Step 32: ['6', '1', '5', '7', '4']
> Step 48: ['6', '1', '7', '5', '9']
> Step 64: ['7', '1', '6', '5', '9']
> ```
>
> Examples with incorrect thinking patterns:
>
> Example 1 (Answer: 8):
> ```
> Question: What is (8 - 2) + 2?
> Answer:
> ```
> Top-5 ranked token decoded from latent thoughts at different thinking steps:
> ```
> Step 16: ['-', '1', '2', '3', '4']
> Step 32: ['6', '4', '1', '-', '5']
> Step 48: ['6', '4', '-', '5', '8']
> Step 64: ['6', '4', '8', '-', '1']
> ```
>
> Example 2 (Answer: 7):
>
> ```
> Question: What is (3 - 2) + 6?
> Answer:
> ```
>
> Top-5 ranked token decoded from latent thoughts at different thinking steps:
>
> ```
> Step 16: ['-', '1', '3', '2', '4']
> Step 32: ['1', '-', '5', '2', '3']
> Step 48: ['3', '4', '5', '2', '1']
> Step 64: ['4', '5', '6', '3', '2']
> ```
> Example 3 (Answer: 7):
>
> ```
> Question: What is (6 - 4) + 5?
> Answer:
> ```
>
> Top-5 ranked tokens decoded from latent thoughts at different thinking steps:
>
> ```
> Step 16: ['-', '1', '3', '2', '4']
> Step 32: ['1', '-', '2', '4', '3']
> Step 48: ['2', '1', '3', '4', '6']
> Step 64: ['1', '2', '3', '5', '4']
> ```
>
> For correct examples, we observe that the correct digit token emerges among the top-k candidates at middle thinking steps, then rises to rank-1 and stabilizes in later steps. In contrast, for incorrect examples, the correct token does not consistently rise in rank, the latent thoughts show fluctuation or drifting behavior and the final step fails to converge to the correct answer. The pattern differences between correct and incorrect latent thoughts demonstrate that the latent reasoning language model encodes meaningful reasoning patterns in its latent thoughts that truly reflect its thinking processes.
>
> ## W3. Ambitious phrases
>
> Thanks for the comment! In Section 6.3, our experimental results demonstrate that LRM can achieve cross-task transferability and therefore hold potential to build a generalist reward model in the latent space. While our empirical results do not claim full transferability across all possible tasks, we believe that they do indicate promising cross-domain potential for future work. We will improve the wording in the revision to make the claims more rigorous.
>
> ## Q2. Why do correct latent trajectories tend to be more compact?
>
> Thanks for the comment! In our experiments we do observe that, in many cases, for the same problem, correct latent trajectories tend to be more concentrated along specific directions while incorrect latent trajectories tend to be more dispersed. Our intuition is as follows:
>
> For correct trajectories, the latent thoughts tend to quickly move toward a stable direction toward the correct solution. Once the model identifies the correct thinking path, subsequent latent thoughts tend to refine this direction instead of exploring alternatives. This produces compact trajectories with early and consistent convergence toward the final latent thought corresponding to the correct answer.
>
> In contrast, incorrect trajectories typically fail to lock onto a coherent solution path. Because the model’s thinking process is uncertain or inconsistent, its latent thoughts continue to drift, backtrack, or explore multiple incompatible directions. This leads to more scattered and less stable trajectories with delayed or inconsistent convergence patterns.

---

### Official Review · Reviewer_Pyxq · 2025-11-01

**Soundness:** 2
**Presentation:** 3
**Contribution:** 2
**Rating:** 4
**Confidence:** 2

**Summary:**

This paper first empirically investigates the property of latent in latent thinking LLMs, such as Huggin-3.5B to understand their internal reasoning processes. They discover that the latent trajectories leading to correct versus incorrect answers are highly distinguishable and provide both visual and quantitative measurements to support this claim. Based on this finding, they successfully train a Latent Reward Model (LRM) that predicts the correctness of the final answer directly based on the sequence of latent thoughts. Finally, the paper proposes Latent Thinking Optimization (LTO), a probabilistic sampling algorithm that leverages the LRM's reward signal to guide the model's generation at test-time. This LTO algorithm is supported by both theoretical justification and strong empirical results, demonstrating performance improvements on math, coding, and general QA tasks.

In general, this paper validates the separation of correct/incorrect latent trajectories and train a classifier LRM that predicts the trajectory correctness sorely based on the latent. Using LRM as reward model to guide the sampling with LTO yields better results than other test-time scaling method.

**Strengths:**

1. Clear motivation and empirical insight: convincingly shows correct vs. incorrect latent trajectories are separable, supported by PCA visualizations and four representation metrics (entropy, effective rank, anisotropy, intrinsic dimension). A lightweight sequence classifier (LRM) can achieve strong predictive power.
2. The sampling algorithm LTO is principled and theoretically sound.
3. Consistent empirical gains on math/coding/QA vs. majority voting, self-correction, and latent heuristics (CoE-R/CoE-C).

**Weaknesses:**

1. Primary Application Targets a Niche Architecture: The paper's methodology is heavily anchored to the Huggin-3.5B model, which features a specific, recurrent "latent thinking" architecture. This model is not a widely adopted or standard foundation model, making it a niche target.
2. Ambiguous Application to General LLMs. The paper's method for applying LTO to standard transformers (like Llama 2 or Mistral) is conceptually ambiguous. In section 6, the author mentioned that "train LRMs using the latent representations from general LLMs". In the appendix D, the authors state "The latent representations of general LLMs from all the layers are regarded as latent chain of thoughts.". Does that implies that hidden states in all layers are concatenated together as input to LRM? Meanwhile, efficiency of such methods on the application to general LLM is not sufficiently discussed.
3. Limited Evaluation Scope for General LLMs: The experiments on general LLMs are constrained to relatively weak benchmarks such as GSM8k, it is unclear if these gains would translate to more capable models on frontier benchmarks.

**Questions:**

1. Can the authors specify how multi-layer latents in general LLM are composed for the LRM as well as the efficiency analysis of LRM on general LLMs?
2. Can the authors provide more experiment results for general LLM on recent benchmarks such as MATH/GPQA-Diamond/AIME?

---

> ### Author Response · Authors · 2025-11-19
>
> We are grateful that reviewer Pyxq thinks that our paper has clear motivation and empirical insight, is theoretically sound, and achieves consistent empirical gains. We sincerely appreciate the questions and would like to offer further clarifications:
>
> ## W1. Primary Application Targets a Niche Architecture
>
> Thanks for the insightful comment. While Huginn-3.5B is a specific recurrent latent-thinking architecture, it is a representative example of a new and important paradigm in which LLMs perform reasoning in latent space rather than relying solely on verbalized chains of thought. Our goal is not to promote this particular model, but to use it as a controlled setting for understanding and improving this emerging latent-thinking paradigm.
>
> Importantly, our contributions are not limited to Huginn-3.5B. We show that both LRM and LTO transfer seamlessly to standard, non-recurrent LLMs, and improve their thinking quality without requiring any architectural modifications. This demonstrates that the proposed methods generalize beyond the niche model and are applicable to general LLMs.
>
> ## W2-1 & Q1-1 Does that imply that hidden states in all layers are concatenated together as input to LRM?
>
> Thanks for the comment! As detailed in Appendix G, for general LLMs, the hidden states from all layers are stacked together to form a sequence of latent thoughts. For example, the hidden representation from layer 1 is treated as latent thought step 1, the representation from layer 2 as step 2, and so on. This stacked sequence of latent thoughts from general LLMs serves as the input to the LRM (a sequence classifier), and it has exactly the same format as the sequences of latent thoughts from Huginn-3.5B.
>
> ## W2-2 & Q1-2 The efficiency analysis of LRM on general LLMs
>
> Thanks for the comment! LRMs are highly efficient both for general LLMs and Huginn-3.5B due to its lightweight architecture with only 2 layers of Transformer encoder. Its training time is small because it only requires a small amount of training data and its inference time is orders of magnitude lower than the inference time of the base LLM. For example, for Llama3-8B, on a single A100 GPU, LRM only requires ~1.5 hrs to train and 6.4GB of memory usage, and its reward computation can be completed within ~0.02s which is negligible compared with the inference time of the base LLM.
> We will better clarify the efficiency of LRMs on general LLMs in the revision.
>
> Training efficiency:
>
> |         |GSM8K|CommonsenseQA|MBPP|MATH|GPQA
> |--------------------------|----------|-----------|--------------------|-----------|--------------------|
> |Total Training Time (h)|1.21|1.53|0.66|1.69|0.42|
> | GPU Memory Usage (GB)| 6.39     |6.40 | 6.39    |6.39 |6.39
>
> Inference efficiency comparison on a single A100 GPU:
>
> |         |GSM8K|CommonsenseQA|MBPP|MATH|GPQA
> |--------------------------|----------|-----------|--------------------|-----------|--------------------|
> |Based Model Inference|8.5s|8.2s|3.0s|12.5s|21.8s
> |Latent Reward Computation| 0.013s     |0.016s | 0.005s    |0.005s |0.005s
>
> ## W3 & Q2 More experiment results for general LLMs on recent benchmarks
>
> Thanks for the comment! Due to limited time and resources, here we evaluate LTO using two more capable LLMs on two more recent, frontier benchmarks. The results show that LTO consistently enhances the reasoning performance across all models and datasets, demonstrating its effectiveness and robustness in improving the latent thinking processes of the general LLMs on recent, frontier benchmarks.
>
> | Model         | Method   | MATH|GPQA|
> |---------------|-----------------------------|------------|-----------|
> | Llama-3-8B    | Base                |0.267 |0.268
> |               | Majority Voting         | 0.335|0.276
> |               | Latent Thinking Optimization|0.375 |0.310
> | Qwen-3-4B   | Base                        | 0.552|0.347
> |               | Majority Voting             | 0.555|0.368
> |               | Latent Thinking Optimization|0.619|0.490

---

### Official Review · Reviewer_aTJi · 2025-11-01

**Soundness:** 3
**Presentation:** 3
**Contribution:** 3
**Rating:** 6
**Confidence:** 4

**Summary:**

This paper investigates latent thinking in reasoning language models, focusing on the Huginn-3.5B model which generates intermediate reasoning steps as latent representations rather than natural language. The authors analyze differences in latent thought trajectories between correct and incorrect answers using visualizations and representation quality metrics, revealing distinguishable patterns. They train a latent classifier as a Latent Reward Model (LRM) to predict correctness from these trajectories and propose Latent Thinking Optimization (LTO), a probabilistic sampling algorithm that optimizes latent policies to favor trajectories likely to yield correct answers. Experiments on math and programming tasks show improvements, and the approach is extended to general LLMs with cross-domain generalization.

**Strengths:**

* The exploration of latent thinking patterns provides novel insights into how LLMs encode reasoning internally, bridging cognitive science inspirations with practical LLM analysis, which could inspire future work on interpretable latent spaces.

* LTO is a computationally efficient alternative to verbal chain-of-thought methods, avoiding overthinking and verbosity, with theoretical grounding in reward optimization and empirical gains on benchmarks like SVAMP and MBPP.

* The generalization of LRM and LTO to standard LLMs (beyond Huginn-3.5B) and across domains is promising, suggesting a scalable, domain-agnostic way to enhance LLM reasoning without heavy natural language generation.

**Weaknesses:**

* The analysis heavily relies on Huginn-3.5B, a specific latent reasoning model; while extensions to general LLMs are claimed, the paper lacks detailed comparisons or ablation on how well LTO performs on diverse LLM model families.

* Evaluation metrics for latent representations (e.g., entropy, anisotropy) are insightful but somewhat indirect; the paper could benefit from more direct interpretability probes or causal interventions to confirm that observed patterns truly reflect "thinking" rather than spurious correlations or memorization artifacts.

* Experimental details on training the LRM (e.g., data scale, hyperparameters) are sparse in the provided sections, and results focus on two datasets; without broader benchmarks or robustness tests (e.g., against adversarial inputs), it's unclear how LTO scales to real-world, noisy reasoning scenarios.

**Questions:**

* How sensitive is LTO to the number of sampled trajectories N or the KL regularization weight β? Did you observe trade-offs between optimization gains and deviation from the reference policy?

* The paper mentions cross-domain generalization of LRM with small training data; what specific domains were tested for transfer (e.g., beyond math/programming to commonsense or scientific reasoning), and how much data was used for fine-tuning in those cases?

---

> ### Author Response · Authors · 2025-11-19
>
> We are grateful that reviewer aTJi thinks that our paper provides novel insights, has both theoretical grounds and empirical gains, and demonstrates generalizability across domains. We sincerely appreciate the questions and would like to offer further clarifications:
>
> ## W1. Comparison on LLM model family
>
> Thanks for the comment! We provide an additional experiment evaluating LTO on the widely-used Llama family. The results show that LTO consistently enhances the reasoning performance across all models, demonstrating its effectiveness in improving the latent thinking processes of the LLM model family.
>
> | Model         | Method   | GSM8K    | GSM-Symbolic | CommonsenseQA       | MBPP       |
> |---------------|-----------------------------|------------|-----------|------------|-----------|
> | Llama-2-7B    | Base                | 0.223 | 0.204  | 0.399     | 0.189     |
> |               | Majority Voting         | 0.275 | 0.302  | 0.493     | 0.193     |
> |               | Latent Thinking Optimization| 0.389| 0.316| 0.606| 0.237|
> | Llama-2-13B   | Base                        | 0.306    | 0.273   | 0.398     | 0.247     |
> |               | Majority Voting             | 0.417      | 0.379   | 0.501     | 0.263     |
> |               | Latent Thinking Optimization| 0.534| 0.442 | 0.650| 0.322|
> | Llama-3-8B  | Base                        | 0.784|0.736|0.742|0.560
> |               | Majority Voting             | 0.801|0.796|0.786|0.570
> |               | Latent Thinking Optimization |0.859|0.821|0.790|0.600
>
>
> ## W2. Interpretability of latent thoughts
>
> Thanks for the comment! Here we provide an interpretability analysis by decoding the latent thoughts at different reasoning steps and examining how they evolve toward (or away from) the correct answer. To make this analysis clear, we use a one-digit arithmetic dataset, where the model must output a single digit answer to an arithmetic question. This setting is ideal for interpretability because the operations are simple and easy to verify, and we can directly inspect how the decoded latent thoughts evolve at the specific token position corresponding to the answer digit. Using the coda module (decoder) from the latent reasoning language model, we decode the latent vector at each thinking step into its top-5 most probable tokens and analyze the progression of latent thoughts. Below we show representative examples with correct and incorrect thinking patterns.
>
> Examples with correct thinking patterns:
>
> Example 1 (Answer: 7):
> ```
> Question: What is (1 * 1) + 6?
> Answer:
> ```
> Top-5 ranked tokens decoded from latent thoughts at different thinking steps:
> ```
> Step 16: ['-', '1', '3', '2', '4']
> Step 32: ['1', '7', '6', '2', '8']
> Step 48: ['7', '6', '1', '8', '2']
> Step 64: ['7', '6', '1', '8', '2']
> ```
>
> Example 2 (Answer: 5):
> ```
> Question: What is (7 + 2) - 4?
> Answer:
> ```
>
> Top-5 ranked token decoded from latent thoughts at different thinking steps:
> ```
> Step 16: ['-', '1', '3', '2', '4']
> Step 32: ['1', '5', '9', '2', '6']
> Step 48: ['5', '1', '-', '7', '4']
> Step 64: ['5', '7', '9', '1', '-']
> ```
> Example 3 (Answer: 7):
> ```
> Question: What is (3 * 2) + 1?
> Answer:
> ```
> Top-5 ranked token decoded from latent thoughts at different thinking steps:
> ```
> Step 16: ['-', '1', '3', '2', '4']
> Step 32: ['6', '1', '5', '7', '4']
> Step 48: ['6', '1', '7', '5', '9']
> Step 64: ['7', '1', '6', '5', '9']
> ```
>
> Examples with incorrect thinking patterns:
>
> Example 1 (Answer: 8):
> ```
> Question: What is (8 - 2) + 2?
> Answer:
> ```
> Top-5 ranked token decoded from latent thoughts at different thinking steps:
> ```
> Step 16: ['-', '1', '2', '3', '4']
> Step 32: ['6', '4', '1', '-', '5']
> Step 48: ['6', '4', '-', '5', '8']
> Step 64: ['6', '4', '8', '-', '1']
> ```
>
> Example 2 (Answer: 7):
>
> ```
> Question: What is (3 - 2) + 6?
> Answer:
> ```
>
> Top-5 ranked token decoded from latent thoughts at different thinking steps:
>
> ```
> Step 16: ['-', '1', '3', '2', '4']
> Step 32: ['1', '-', '5', '2', '3']
> Step 48: ['3', '4', '5', '2', '1']
> Step 64: ['4', '5', '6', '3', '2']
> ```
> Example 3 (Answer: 7):
>
> ```
> Question: What is (6 - 4) + 5?
> Answer:
> ```
>
> Top-5 ranked tokens decoded from latent thoughts at different thinking steps:
>
> ```
> Step 16: ['-', '1', '3', '2', '4']
> Step 32: ['1', '-', '2', '4', '3']
> Step 48: ['2', '1', '3', '4', '6']
> Step 64: ['1', '2', '3', '5', '4']
> ```
>
> For correct examples, we observe that the correct digit token emerges among the top-k candidates at middle thinking steps, then rises to rank-1 and stabilizes in later steps. In contrast, for incorrect examples, the correct token does not consistently rise in rank, the latent thoughts show fluctuation or drifting behavior and the final step fails to converge to the correct answer. The pattern differences between correct and incorrect latent thoughts demonstrate that the latent thoughts encode meaningful reasoning patterns that truly reflect the thinking processes, rather than spurious correlations or memorization artifacts.

---

> ### Author Response · Authors · 2025-11-19
>
> ## W3-1. Experimental details on LRM
>
> As mentioned in Appendix E, the LRM is 2-layer Transformer with Sinusoidal positional encoding to encode the sequence of latent thoughts. The configuration of the latent classifier follows the configuration of the language model it is trained for (e.g., if the LLM has hidden dimensionality 5280, number of attention heads 55, and MLP hidden size 17920, then the LRM also has the same hidden dimensionality, attention heads and MLP hidden size). Training is performed with binary cross-entropy loss for 10 epochs using the Adam optimizer with a learning rate of 5e − 6.
>
> As mentioned in Appendix G2, for GSM-8K, SVAMP, and CommonsenseQA, we sample 5 different latent thinking trajectories and answers per problem from the training split, resulting in 37635/176905/48705 data samples for GSM-8K, SVAMP, and CommonsenseQA respectively. For MBPP, which contains only 373 training problems, we sample 50 latent thinking trajectories and answers per problem to ensure sufficient training data, resulting in 18650 training data. Such data scale is considerably smaller than the data samples (~800k) required to train a language-based process reward model [1], demonstrating that LRM can be efficiently trained with only a small amount of data.
>
> Reference
>
> [1] Lightman, Hunter, et al. Let's verify step by step. The Twelfth International Conference on Learning Representations. 2023.
>
> ## W3-2. Results focus on two datasets, Evaluation on broader benchmarks.
>
> Thanks for the comment! We respectfully disagree that our evaluation results are not focused on just two datasets. In Section 6.2, we have already evaluated the performance of LTO on **four** general LLMs using **five** benchmarks from diverse reasoning settings. To further demonstrate the effectiveness of robustness of LTO, here we provide an additional analysis on two broader benchmarks, MATH and GPQA, which are known to be relatively noisy and pose more challenging reasoning conditions. The results show that LTO consistently enhances the reasoning performance across all models and datasets, demonstrating its effectiveness and robustness in improving the latent thinking processes of general LLMs on broader benchmarks under noisy and challenging conditions.
>
> | Model | Method| MATH|GPQA|
> |----------|---------|-----------|--------|
> | Llama-3-8B    | Base|0.267|0.268
> |       | Majority Voting |0.335|0.276
> |       | Latent Thinking Optimization|0.375|0.310
> | Qwen-3-4B   | Base  |0.552|0.347
> |       | Majority Voting  |0.555|0.368
> |       | Latent Thinking Optimization|0.619|0.490
>
> ## Q1-1. Sensitivity to the sampled trajectories N
>
> As shown in Appendix H2 and Figure A4, performance steadily improves as N increases. Even with a very small budget (e.g., N = 2), LTO still achieves substantial performance improvement compared with the base model, demonstrating that LTO is sample-efficient without the need for a large sampling budget.
>
> ## Q1-2. Sensitivity to the KL regularization weight beta
>
> As shown in Appendix H3 and Figure A5, LTO consistently outperforms the base model with different KL regularization weight beta, demonstrating that it can reliably improve the latent thinking processes with different choices of the hyperparameter.
>
> ## Q1-3. Trade-offs between optimization gains and deviation from the reference policy
>
> The KL-regularization weight beta ensures the optimized policy will not collapse into a degenerated policy that significantly deviates from the base policy. A small beta puts more weight on the latent trajectories with higher reward, while a larger beta ensures the policy stays closer to the base policy with less deviation. Moreover, if we remove beta (set beta→0), LTO will reduce to best-of-n sampling in latent space, i.e., we sample N latent trajectories, score them with the LRM, and pick the highest-scoring one. This strategy only works well if the LRM is nearly perfect with almost 1.0 classification accuracy for correct and incorrect latent trajectories. In practice, the LRM is learned and not perfect. Without any regularization, LTO may exploit the LRM’s errors and select suboptimal latent thoughts. By including a modest beta value, we constrain the optimized policy so that it does not drift too far from the base policy, thereby preserving the diversity of sampled latent thoughts and mitigating overfitting to reward model noise.
>
> In our experiments, we also observe this trade-off with different choices of betas. When we push beta→0, performance consistently drops across different tasks. In contrast, tuning beta to a modest nonzero value yields better results. Setting beta to a very large value also degrades performance, since the optimized policy will not be very different from the base policy. Although our setting is test-time scaling where the policy deviation may not be as large as in full RL fine-tuning, we still need a modest beta value to achieve the optimal performance.

---

> ### Author Response · Authors · 2025-11-19
>
> ## Q2-1. Domains tested for Transfer
>
> As shown in Section 6.3, we evaluate the transfer-learning capability of the LRMs across five datasets: GSM8K, GSM-Symbolic, SVAMP (math), CommonsenseQA (commonsense reasoning), and MBPP (coding). For every pair of datasets, we conduct cross-domain evaluations, resulting in six transfer settings: math → commonsense reasoning, math → coding, commonsense reasoning → coding, commonsense reasoning → math, coding → math, and coding → commonsense reasoning. This allows us to comprehensively assess the LRM’s ability to generalize across different task domains.
>
> ## Q2-2. Data required to train the cross-domain LRM
>
> Thanks for the comment! As mentioned in Seciton 6.3, our cross domain experiments test the transferability of both dataset-specific LRMs and a cross-domain general LRM. The dataset-specific LRMs are trained on one source dataset and evaluated on the target dataset (e.g., train with 37,635 data samples from the training split of GSM8K, then evaluate on GSM-Symbolic SVAMP, CommonsenseQA, and MBPP respectively, and similary train with SVAMP/CommonsenseQA/MBPP, then evaluate on all other datasets)  without any finetuning using the target dataset. To train the cross-domain general LRM, we merge the training splits from all datasets (37,635 from GSM8K, 176,905 from SVAMP, 48,705 from CommonsenseQA, and 18,650 from MBPP), yielding a combined pool of about 282K training data. This data size is relatively modest, yet it is sufficient for training a strong cross-domain general LRM.

---

### Author Response · Authors · 2025-11-27
**Looking forward to discussion and feedback**

Dear reviewers:

We sincerely appreciate your insightful comments and believe we have addressed them. As the author response period will close soon, please feel free to reach out if you have any further questions or comments. We would be more than happy to address them.

Thanks, authors of Latent Thinking Optimization

---

> ### Author Response · Authors · 2025-11-29
> **Summary of Revision**
>
> We sincerely thank the reviewers for their constructive feedback and valuable suggestions.
>
> We are grateful that the reviewers think that our paper provides novel insights into how LLMs encode reasoning in the latent space (reviewer aTJi, Pyxq) with convincing quantitative and qualitative analysis on latent thoughts (reviewer Pyxq, jnr6, qGuA), the proposed LTO has both theoretical guarantee and empirical gains (reviewer aTJi, Pyxq, jnr6, qGuA), and demonstrates broader utility on general LLMs and transferability across diverse tasks (reviewer aTJi, jnr6, qGuA).
>
> To address the reviewers’ concerns, in our rebuttal, we have made the following revisions:
>
> (1) We provide an interpretability analysis on the latent thoughts to provide more insights into how the model thinks in the latent space in response to reviewer aTJi and jnr6.
>
> (2) We provide additional experimental results to evaluate LTO on broader, more challenging benchmarks with a diverse LLM family to demonstrate its effectiveness and robustness in improving the latent thinking processes of the general LLMs under noisy and challenging conditions in response to reviewer aTJi and Pyxq.
>
> (3) We provide an additional efficiency analysis on general LLMs in response to reviewer Pyxq.
>
> (3) We clarify the necessity of the KL regularization in response to reviewer aTJi and jnr6.
>
> (4) We clarify the experimental details (data scale, hyperparameters and data format) on training the latent reward models on general LLMs in response to reviewer aTJi and Pyxq.
>
> (5) We clarify the formulation of the optimization problem for LTO in response to reviewer jnr6.
>
> (6) We clarify that the latent representations are scale-invariant in response to reviewer qGuA.
>
> We have also provided detailed responses to answer other questions from the reviewers, including the sensitivity to the sampled trajectories N (reviewer aTJi), domains and data for cross-domain transfer (aTJi), the scope and application target of our paper (reviewer Pyxq), clarification of phrases in Section 6.3 (reviewer jnr6), the reason why correct latent trajectories tend to be more compact (reviewer jnr6), clarification of the controlled experiment on early-stopped latent representations (reviewer qGuA), future directions for non-binary reward signals (reviewer qGuA), and usage of large language models (reviewer qGuA).
>
> We deeply appreciate the reviewers’ constructive feedback and have carefully revised the paper to address each concern. We would like to extend our sincere gratitude again to the reviewers for their valuable comments and insights, which have grealty helped us to further improve the quality of our work.

---

### Meta-Review · Area_Chair_S91M · 2025-12-19

**Summary:**

While initial concerns were raised regarding the experimental design and the interpretation of the latent thinking process, the authors have substantially addressed these points in their rebuttal. They have supplemented their work with additional experiments using more reasoning models across several benchmark datasets and have provided clarifying intermediate results. Overall, this is a borderline paper above the acceptance threshold.

**Reviewer Concerns:**

The major concerns raised by the reviewers include 1. The initial lack of explanation regarding how latent thinking functions in practice was addressed by the authors' provision of intermediate results in their rebuttal. 2. The original experiments, confined to the Huginn-3.5B model, raised questions about generalizability. The authors have now strengthened this aspect by extending their evaluation to include LLaMA-family models. Additional concerns about specific methodological components and experimental settings were also satisfactorily resolved through supplemental experiments provided during the rebuttal phase.

**Reviewer Scores:**

As the concerns raised by the reviewers are not very critical, they have been empirically addressed in the rebuttal. I think Pyxq may increase his/her rating to 6. For the 3 reviewers, they would maintain or slightly increase their ratings.

---

### Decision · Program_Chairs · 2026-01-26

Accept (Poster)